# Copper Coordination Compounds as Biologically Active Agents

**DOI:** 10.3390/ijms21113965

**Published:** 2020-05-31

**Authors:** Olga Krasnovskaya, Alexey Naumov, Dmitry Guk, Peter Gorelkin, Alexander Erofeev, Elena Beloglazkina, Alexander Majouga

**Affiliations:** 1Chemistry Department, Lomonosov Moscow State University, Leninskie gory 1,3, 119991 Moscow, Russia; asselassel72@yandex.ru (A.N.); dmh200949@gmail.com (D.G.); Erofeev@polly.phys.msu.ru (A.E.); beloglazki@mail.ru (E.B.); alexander.majouga@gmail.com (A.M.); 2Department of Materials Science of Semiconductors and Dielectrics, National University of Science and Technology (MISIS), Leninskiy prospect 4, 101000 Moscow, Russia; peter.gorelkin@gmail.com; 3Mendeleev University of Chemical Technology of Russia, Miusskaya Ploshchad’ 9, 125047 Moscow, Russia

**Keywords:** copper coordination compounds, antitumor drug, antibacterial agents, PET imagining agents, mycobacterium tuberculosis, Alzheimer’s disease

## Abstract

Copper-containing coordination compounds attract wide attention due to the redox activity and biogenicity of copper ions, providing multiple pathways of biological activity. The pharmacological properties of metal complexes can be fine-tuned by varying the nature of the ligand and donor atoms. Copper-containing coordination compounds are effective antitumor agents, constituting a less expensive and safer alternative to classical platinum-containing chemotherapy, and are also effective as antimicrobial, antituberculosis, antimalarial, antifugal, and anti-inflammatory drugs. ^64^Cu-labeled coordination compounds are promising PET imaging agents for diagnosing malignant pathologies, including head and neck cancer, as well as the hallmark of Alzheimer’s disease amyloid-β (Aβ). In this review article, we summarize different strategies for possible use of coordination compounds in the treatment and diagnosis of various diseases, and also various studies of the mechanisms of antitumor and antimicrobial action.

## 1. Introduction

Metal-containing therapeutic agents comprise a fundamental class of drugs for treating tumors. Although many metal-containing drugs based on gold, ruthenium, gallium, titanium, and iron are in preclinical and clinical trials phases I and II [1], cisplatin and also second- and third-generation platinum coordination compounds (carboplatin, oxalyplatin, and picoplatin) are still the most effective antitumor agents used in clinical practice [2]. The clinical use of platinum-based drugs entails many severe side effects, such as nephrotoxicity [3], neurotoxicity [4], and also ototoxicity and myelosuppression [5].

It is assumed that antitumor drugs based on endogenous metals (Co, Cu, Zn, and Fe) are less toxic as compared with platinum analogues [6]. Copper-containing coordination compounds were found to be promising antitumor therapeutic agents that act by various mechanisms such as inhibition of proteasome activity [7,8], telomerase activity [9], reactive oxygen species (ROS) formation [10,11], DNA degradation [12], DNA intercalation [13], paraptosis [14], and others. 

Copper is an element of fundamental importance for the formation and functioning of several enzymes and proteins, such as cytochrome C oxidase and Cu/Zn superoxide dismutase, which are involved in the processes of respiration, energy metabolism, and DNA synthesis [15]. Most Cu(II) coordination compounds quickly form adducts with glutathione in the cell medium, which leads to the formation of a coordination compound of monovalent Cu(I) capable of generating a superoxide anion, which can induce ROS formation in a fenton-like reaction [16]. Due to high redox activity, the therapeutic efficacy of copper coordination compounds is not limited to antiproliferative action. Copper coordination compounds can be highly effective in treating viral infections [17], inflammatory diseases [18], and microbial infections [19] by multiple mechanisms of action. A Cu(II) coordination compound based on indomethacin is currently used in veterinary practice as an anti-inflammatory drug [20]. 

Malignant and inflamed tissues metabolize an increased amount of copper as compared with healthy tissues [21], which gives copper-containing coordination compounds an additional advantage over other metal-containing drugs. Various delivery systems for copper-based therapeutic agents and also for copper and chelating ligand separate delivery have been developed to enhance their delivery into tumor tissues [22,23]. 

Development of novel copper coordination compounds with antitumor activity is a promising and relevant area of medical chemistry [24,25,26,27]. A number of copper/Disulfiram-based drug combinations are in recruiting clinical trials as inexpensive and highly effective antitumor agents for metastatic breast cancer therapy and as diagnostic tools [28]. Their clinical success has triggered the development of delivery and controlled release systems for copper coordination compounds, as well as a search for novel copper-containing anticancer agents [29,30]. A brief and clear summary of promising in vivo anticancer activity of this type of drugs, along with relevant and current clinical trials was reported by Tabti1 et al. [31], but rapid development of copper-based therapy has caused rapid changes in clinical data. In a search for methods to overcome the poor water solubility of copper complexes, Wehbe et al. briefly summarized the use of copper complexes as antineoplastic agents [32]. However, after the publication of a detailed high-quality review by Santini et al. [33], a small number of works examined, in detail, the latest biological aspects of the use of copper complexes as therapeutic agents. Recently, Ong et al. reported a metal application in tropical diseases treatment, expanding the understanding of the applications of copper-containing coordination compounds [34]. 

In this review we provide a summary of different publications of recent years, paying attention to the variety of biological studies on the therapeutic and diagnostic potential of copper coordination compounds. We have emphasized the most abundant methods used to assess the mechanism of antitumor action and other therapeutic effects. This review could be useful to researchers engaged in medicinal application of copper-containing agents, affecting various uses of copper coordination compounds such as anticancer, antituberculosis, antimicrobial, anti-inflammatory, antibacterial agents, as well as PET-imaging agents for the diagnosis of malignant neoplasms and Alzheimer’s disease.

## 2. Copper Coordination Compounds Based on Ligands with Various Donor Atoms

### 2.1. N- and O-Donor Ligands

Casiopeínas comprise a family of copper coordination compounds with promising results for treating colorectal cancer and acute myeloid leukemia. Several Casiopeínas compounds have shown significant therapeutic efficacy, and two of them, Casiopeina III-ia 1 and Casiopeina II Gly 2 (Figure 1), have underwent a number of clinical trials as drugs for the treatment of leukemia [35].

Several hypotheses have been developed regarding the mechanism of action of Casiopeinas, including ROS formation, phosphate hydrolysis, DNA damage, and DNA intercalation [36]. In addition, one of the latest studies [37] of coordination compounds of the Casiopeínas class has shown an antiproliferative effect in *Giardia intestinalis* trophozoite cultures, a pathogen causing an infectious disease that affects residents of developing countries. The antiproliferative effect of coordination compounds is explained by their ability to interact with the cell membrane and increase the ROS concentration in the parasite from the first hours of exposure (2–6 h). It was found that these compounds caused the death of trophozoite cells as a result of apoptosis. Guillermo de Anda-Jáuregui et al. recently constructed a network with deregulated biological pathways featuring links between pathways that crosstalk with each other. Through this approach, the following three features of Casiopeina treatment were identified: (a) perturbation of signaling pathways related to apoptosis induction, (b) perturbation of metabolic pathways, and (c) activation of immune responses [38].

Copper coordination compounds **3**–**5** with Schiff bases as ligands were obtained by condensation of 5-dimethylcyclohexane-1,3-dione and a hydrazine derivative by Shoair et al. [39] (Figure 2).

Coordination compounds **3**–**5** showed the ability to intercalate calf thymus DNA and also showed cytotoxic activity on the cell lines of liver cancer HepG-2 (human liver cancer cell line of hepatocellular carcinoma) and breast cancer MCF-7 (breast cancer cell line of invasive breast ductal carcinoma) (Table 1). The toxicity of the ligands and their corresponding coordination compounds is comparable. Complex **4** showed the greatest cytotoxic activity on MCF-7 cell lines.

The antimicrobial activity of ligands **L3**–**L5** and Cu(II) complexes **3**–**5** were tested against bacteria and fungi. All ligands and complexes were found to have antibacterial activity against Gram-negative *Escherichia coli* (except **3**), Gram-positive *Staphylococcus aureus*, and *Candida albicans* (Table 2).

Copper-containing antitumor agents, with Schiff-base ligands based on hydrazone with a pyridine coligand, were investigated by QingYou Mo et al. [40]. Introducing N-containing coligands such as imidazole, pyridine, quinoline, phenanthroline, and their derivatives can affect the hydrophobicity, the geometry of the coordination compound, and consequently, the antitumor activity. Copper coordination compounds **6**–**8** with Schiff-base ligands and also coordination compounds **9**–**11** containing pyridine as a ligand were obtained (Figure 3). Studies of antiproliferative activity showed that introducing a pyridine coligand into the structure of the coordination compound increases cytotoxic activity as expected. Coordination compounds **9**–**11** containing a pyridine coligand exhibit antiproliferative activity in vitro with an IC_50_ ranging from 1.12 to 6.31 μM (MCF-7 breast cancer cells), while non-coligand analogues **6**–**8** have an IC_50_ in the range from 3.66 to 18.61 µM (MCF-7 breast cancer cell line of invasive breast ductal carcinoma).

For coordination compounds **9**–**11**, cytotoxicity studies were also conducted on cisplatin-resistant lung cancer cell lines (A549cisR cisplatin-resistant lung cancer cell line of adenocarcinomic human alveolar basal epithelial cells). High toxicity was shown with an IC_50_ in the range of 3.77 to 6.03 μM (IC_50_ > 50 μM for cisplatin). Studies of the mechanism of the cytotoxic effect of coordination compound **11** showed that the drug causes DNA degradation, which triggers the mechanism of ROS-mediated apoptosis of mitochondrial dysfunction.

D. Anu et al. reported tetra-nuclear mixed-valence copper (I/II) coordination compound **12** with promising antitumor activity [41] (Figure 4). The structure of the obtained coordination compound was confirmed by X-ray diffraction.

A spectrophotometric study of DNA intercalation by ligand **L12** and coordination compound **12** was performed by titration of a calf thymus DNA solution with the solutions of coordination compound **12**. The binding constants of **L12** and coordination compound **12** were, respectively, (2.34 ± 0.60) × 10^5^ M^−1^ and (3.50 ± 0.73) × 10^5^ M^−1^, indicating a weak interaction with the secondary structure of DNA. BSA protein binding was confirmed by fluorescence titration. The antioxidant activity of coordination compound **12** was proven by the ability to decrease the reduction of Mo(VI) to Mo(V) and by, subsequent, formation of a complex at acidic pH. The ability of **L12** and coordination compound **12** to cause apoptosis in MCF-7 cells was proven by the acridine orange/ethidium bromide (AO/EtBr) staining method.

Studies of antiproliferative activity on breast cancer cells MCF-7 and lung cancer cells A-549 showed an IC_50_ of 32 ± 1.0 μM (MCF-7 breast cancer cell line of invasive breast ductal carcinoma), 15 ± 1.5 μM (A-549 lung cancer cell line of adenocarcinomic human alveolar basal epithelial) for ligand **L12** and 25 ± 1.0 μM (MCF-7) and 12 ± 1.0 μM (A-549) for coordination compound **12**.

### 2.2. N- and S-Donor Ligands

Elesclomol is an injectable chemotherapeutic agent ligand **L13** with low molecular weight (Figure 5), which demonstrated clinical efficacy in acute myeloid leukemia [42]. This drug is also at the first stage of clinical trials as a therapeutic agent for leukemia [43]. Elesclomol has been proven to exert an antitumor effect by forming a Cu(II) coordination compound in situ, and the corresponding coordination compound causes oxidative stress inside the malignant cell. The redox reaction Cu(II)/Cu(I) disrupts mitochondrial respiration and causes ROS formation. Ultimately, coordination of copper with the elesclomol ligand disrupts the production and metabolism of cellular energy and triggers the path of mitochondrial apoptosis in tumor cells, leading to their death [44].

An antitumor activity of the redox-active copper coordination compound was also confirmed in [45], where the potential effectiveness of elesclomol in treating ovarian cancer was shown. The elesclomol ligand showed antiproliferative activity on six cell lines of gynecological cancer with an IC_50_ of 0.173 µM and an IC90 of 0.283 µM (tests were performed with Cu-preincubated cell lines).

Due to redox properties, copper coordination compounds not only are effective redox-active antitumor agents but also are effective in treating bacterial and fungal infections. Tuberculosis (TB) caused by *Mycobacterium tuberculosis* (Mtb) is an infection causing more deaths than acquired immunodeficiency syndrome. First-line drugs, such as rifampicin, successfully coped with bacterial pneumonia, but drug resistance requires seeking new chemotherapeutic agents. A new triple-drug combination for treating TB is a combination of oxidant [46] and redox-active drugs [47] coupled with a third drug with a different mode of action. Therefore, the redox activity of copper ions coupled with the fact that the immune system uses copper to eliminate bacterial infections makes copper coordination compounds promising antibacterial, and in particular, antituberculosis chemotherapeutic agents.

Recent studies by Ngwane et al. [48] demonstrated that elesclomol is relatively potent against Mtb H37Rv with a minimum inhibitory concentration of 10 μM (4 mg/L). In addition, against multidrug resistant clinical isolates of Mtb, it displays additive interactions with known tuberculosis drugs such as isoniazid and ethambutol, and a synergistic interaction with rifampicin.

Controlled supplementation of elesclomol with copper leading to the formation of compound **13** in culture medium increased Mtb sensitivity by >65-fold. (Table 3)

Cu-ATSM **14** is a biologically active copper coordination compound based on thiosemicarbazides (Figure 6). This drug labeled with the radioactive isotopes ^64^Cu, ^62^Cu, and ^60^Cu was used as a PET hypoxia imaging agent in head and neck cancer [49]. It demonstrated better results in clinical trials than the 18-fluorodeoxyglucose used in clinical practice [50,51].

Drug accumulation in hypoxic areas is associated with redox transitions of Cu(II)/Cu(I). Labeled with a radionuclide tag, Cu-ATSM **14** penetrated into cells by passive diffusion and underwent glutathione reduction. Under normoxic conditions, the labile coordination compound Cu(I) is oxidized by intracellular oxygen to the coordination compound Cu(II) and leaves the cell. In contrast, under hypoxic conditions, the Cu(I) coordination compound dissociates into a ligand and a metal ion, which binds to intracellular chaperone proteins leading to the accumulation of a radionuclide in hypoxic regions of tumors. The oxidation process of Cu(I) is so fast that a noticeable intracellular reduction of Cu(II) ATSM occurs only in hypoxic (tumor) cells, while the drug leaves healthy cells without any changes [52] (Figure 6).

Coordination Cu-ATSM **14** also proved to be an effective drug capable of slowing the progression of amyotrophic lateral sclerosis (ALS) disease and improving the respiratory and cognitive function of patients. Currently, the drug Cu(II)-ATSM is undergoing clinical trials as a drug for the treatment of ALS. Patient registration for phase III clinical trials of the treatment of this disease began in November 2019 in Australia [53].

Anjum et al. recently reported eight thiosemicarbazido-based Cu(II) complexes, Cu-ATSM analogues **15**–**22**, with promising antitumor activity [54] (Figure 7).

The toxicity of Cu(II) complexes **15**–**22** could be decreased by co-incubation with the nontoxic Cu chelator tetramolibdate (TM) or the antioxidant N-acetylcysteine (NAC), suggesting a mechanism of Cu-induced oxidative stress. The redox behavior of Cu(II) complexes was also of interest. Depending on electron-donating effects of the di-substitutions on the diimine backbone, the Cu(II/I) redox potential itself was changed, and the cytotoxicity changed as a result. The Cu(II/I) redox potential was also proposed to govern the hypoxia selectivity of coordination compounds, but no selective toxicity under hypoxic conditions was shown.

The ability of copper coordination compounds to successfully penetrate the blood-brain barrier has inspired some researchers to create copper-based preparations for visualizing pathological changes in Alzheimer’s disease. One of the major pathological hallmarks of the disease is the presence of extracellular senile plaques in the brain, consisting of an insoluble aggregated peptide called amyloid-β (Aβ), a 39−43 amino acid peptide [55]. Clinically used derivatives of benzothiazole, stilbene, and stripylpyridine labeled with 18-fluorine or 11-carbon are used for PET imaging of plaques by binding to a hydrophobic pocket of the peptide [56,57]. In addition, Zn^2+^ and Cu^2+^ cations have been proven to promote aggregation of amyloid plaques, which provides them with an advantage in binding to amyloid due to the increased consumption of these metals by amyloids [58,59].

A standard approach in developing Aβ PET imaging drugs is to modify a Cu-ATCM drug with a benzothiazole/stilbene moiety, which ensures the binding of the drug to the amyloid plaque. This developmental approach has been used by some Australian researchers (Figure 8).

Hickey et al. [60] succeeded in designing a copper radiopharmaceutical Cu(II)-ATSM with an appended stilbene functional group for Aβ plaque imaging. Binding of compounds **24** and **25** (coordination compound **23** was quite insoluble) to Aβ plaques was clearly evident as demonstrated by epi-fluorescent microscopy. Aβ-specific 1E8 antibody was used as a control.

The biodistribution of coordination compounds **24** and **25** radiolabeled with ^64^Cu in wild-type mice after intravenous tail injection (∼13MBq) displayed good brain uptake of coordination compound **25** (1.11% ID/g) at 2 min after injection, dropping to 0.38% ID/g at 30 min. This indicates that coordination compound **25** can rapidly cross the blood-brain barrier of normal mice with a highly desirable fast washout from the brain as anticipated with no Aβ plaques to trap the imaging agent. Micro-PET images of pre-injected wild-type mice were also obtained.

The biodistribution of coordination compounds **27**–**30** radiolabeled with ^64^Cu in wild-type mice showed the best brain uptake results for coordination compound **30** (1.54% ID/g at 2 min after injection, dropping to 0.77% ID/g at 30 min). TEM images of Aβ1−42 model fibrils treated with compound **28** or **30** demonstrated dramatic changes in the structural morphology.

An alternative methodology is based on elemental mapping using laser ablation inductively coupled plasma mass spectrometry LA-ICP-MS. A sample of nonradioactive isotopically enriched ^65^Cu-**30** was used. Coordination compound **24** was used as a control. The benzofuran containing complex ^65^Cu-**30** appeared to bind with improved differentiation when compared with the styryl-pyridine containing complex ^65^Cu-**24** and potentially offered better sensitivity for amyloid.

On the basis of these results, radiolabeled copper coordination compound could be used to assess amyloid pathology in AD patients using PET. The redox properties of copper ions, the ability to reduce intracellularly, selective accumulation in hypoxic areas, blood-brain barrier penetration, and stability in a blood flow provides copper-containing therapeutic agents features for use as not only therapeutic but also diagnostic and theranostic agents.

### 2.3. N/N-Donor Ligands

Because pathogens with multidrug resistance are emerging and new effective antibiotics against them are lacking, metal-containing coordination compounds have become of interest as antibacterial agents. The effectiveness of copper coordination compounds in the treatment of bacterial and fungal infections was mentioned above (coordination compounds **3** and **5** with antibacterial and antifungal activity [39] and Escimolol-based coordination compound **13** in Mtb treatment [48]). A copper-based coordination compound **31** with antimalarial activity against *Plasmodium falciparum* was developed by [63] (Figure 9). The antimalarial activities in vitro of compound **31** and its ligand were respectively estimated as ED50 = 0.13 and >30 mg / ml for coordination compound **31** and ligand **L31**.

Beeton et al. reported nine copper coordination compounds **32**–**39** based on 1,10-phenanthroline and also their platinum and palladium analogues compounds **40** and **41** with antimicrobial and antibiofilm activity [64] (Figure 10).

The resulting coordination compounds showed higher antimicrobial activity as compared with free ligands against Gram-positive and Gram-negative bacterial strains and also increased antibiotic activity as compared with the standard preparation vancomycin against the clinical strain of methicillin-resistant *Staphylococcus aureus* (MRSA) (Table 4).

The authors associated the action mechanism of coordination compounds **32**–**41** with interactions with the bacterial chromosome, which led to a decrease in bacterial reproduction. The redox activity of copper ions is dependent on the presence of reducing agents. This thiol is glutathione in most Gram-negative bacteria and bacillithiol in several Gram-positive bacteria [65]. Coordination compounds **40** and **41** based on Pt and Pd do not show significant antimicrobial activity, which also indicates that the antibacterial activity is associated with the redox/nuclease activity of copper ions.

Cu(II) coordination compounds **32**–**41** are less active than vancomycin on planktonic cells (Table 4) and are relatively much more active on biofilms. Copper-induced DNA damage can lead to death irrespective of the physiological state or growth rate of the bacterial cells.

Hence, copper coordination compounds can not only be effective synthetic antibacterial agents but also more effective as compared with classical antibiotic therapy because of their mechanism of action.

Brandão et al. reported Cu(II) coordination compounds based on thiochrome, the oxidized form of vitamin B1, with promising cytotoxic activity [66]. The ligand was obtained by oxidizing thiamine with copper (II) chloride. The resulting coordination compounds crystallize in the form of two structures, compounds **42** and **43** (Figure 11). Biological studies on human colon adenocarcinoma cells Caco-2 showed that both compounds reduce the viability of these cells more than thiamine or thiochrome. To investigate the mechanism responsible for the cytotoxic effect of compounds **42** and **43**, the authors tested the alleged involvement of changes in oxidative stress levels. By adding N-acetylcysteine, they ruled out ROS formation, which ultimately showed no change in the cytotoxic effect of compound **42** and only a slight decrease in the cytotoxicity of compound **43**. Therefore, oxidative stress does not seem to play an important role in the mechanism of biological action of these compounds, which indicates that the ability to generate ROS is important but is not always the main mechanism of the cytotoxic action of copper coordination compounds. A comparison of the cytotoxic activity of these compounds on cell lines (IC_50_ = 146 μM for compound **42** and IC_50_ = 191 μM for compound **43**) with cisplatin (IC_50_ = 274 μM) shows that these copper coordination compounds are more effective on human colon adenocarcinoma cell line of heterogeneous human epithelial colorectal adenocarcinoma Caco-2.

Krasnovskaya et al. [67] reported a Cu(II) coordination compound **44** based on 2-aminoimidazolone with promising antitumor activity (Figure 12). Coordination compound **44** showed antiproliferative activity higher than cisplatin (IC_50_ MCF-7 13.67 ± 0.81 μM).

Three copper complexes with potential anticancer and nonsteroidal anti-inflammatory activity were reported by Hussain et al. [68] (Figure 13). Coordination compounds **45**–**47** with benzimidazole-derived scaffolds were synthesized in accordance with the following scheme. In addition to antitumor activity, the compounds were proposed as potential candidates for NSAIDs.

Human serum albumin (HAS) binding of compounds **45***–***47** was evaluated using HAS fluorescence quenching in the presence of coordination compounds. The results showed that the *K*_SV_ values (slope of the Stern–Volmer plots) were of the order of 10^5^, thus indicating strong quenching.

An Annexin/FITC assay showed that the three complexes **45***–***47** exhibited an increase in apoptotic cells to a significant level followed by necrosis. Glutathione depletion along with ROS formation in MCF-7 cells after treatment with coordination compounds **45***–***47** was also shown. The interaction of complexes with COX-2 inhibitor was also confirmed, which can be a mechanism of action of these potential NSAIDs. Coordination compounds **45***–***47** were tested in vivo on albino rats and mice for anti-inflammatory, antipyretic, and analgesic activities. The results showed that **45** and **47** have significant dose-dependent anti-inflammatory and analgesic activities at a lower concentration.

Sliwa et al. reported synthesis, characterization, and biological activity of three water-soluble copper(II) complexes [Cu(NO_3_)(PTA=O)(dmphen)][PF_6_] **48**, [Cu(Cl)(dmphen)2][PF_6_] **49**, and (Cu(NO_3_)_2_(dmphen)) **50** [69] (Figure 14).

The cytotoxic activity of compounds **48**–**50** was evaluated on the normal human dermal fibroblast (NHDF), human lung carcinoma (A549), epithelioid cervix carcinoma (HeLa), colon cancer cell line of supraclavicular lymph node metastasis (LoVo), and breast cancer cell line of invasive breast ductal carcinoma (MCF-7) cell lines (Table 5). All coordination compounds were more active than cisplatin but, expectedly, showed no significant selectivity to healthy cells. The interaction of compounds **48***–***50** with human apo-transferrin, causing a conformational change of the protein, was also proved using fluorescence and circular dichroism spectroscopy.

Milani et al. reported seven novel Cu(I) complexes **51**–**57** with potent antiproliferative activity for human primary glioblastoma cell line U87cells [70] (Figure 15 and Table 6).

Flow cytometry was conducted for accessing the apoptosis rate of U87 cells treated with compounds **51***–***57** after 24 h. All the complexes, except compound **54**, significantly induced apoptosis in U87 cells. The ability of compounds **51***–***57** to inhibit cancer cell growth by induction of cell cycle arrest was also estimated. Treatment of U87 cells with compounds **51**–**57** caused a marked arrest of G1, a growth phase that plays a key role in cell cycle progression and ensures that DNA is ready for synthesis. To validate the hypothesis that **51***–***57** triggered apoptosis in treated U87 cells by the effect on the expression level of apoptotic and anti-apoptotic genes, an RT-PCR assay was conducted. The results obtained showed an increased level of caspase-independent apoptosis genes including P53, P21, Bid, and Bax in U87 cells after treatment by all the complexes except compounds **54** and **55**. The level of anti-apoptotic genes Bcl-2 and Bcl-xL was markedly inhibited in the presence of compounds **55** and **56**.

The investigation of the antitumor activity of Cu(I) coordination compounds **51**–**57**, thus, demonstrated an inhibition of cell growth, cell cycle progression, migration ability, and expression level of anti-apoptotic genes and an induced apoptosis, necrosis, and expression level of apoptotic genes in a dose- and time-dependent manner in treated U87 cells.

Kacar et al. reported coordination compound **58** based on a pyridyl-fluorobenzimidazole scaffold [71] (Figure 16).

Compound **58** showed antiproliferative and apoptotic effects on NIH/3T3 normal fibroblast cells and on SPC212 mesothelioma and DU145 prostate cancer cells. The most significant IC_50_ values were found against DU145, i.e., 37.0, 21.1, and 10.0 mM for the 24, 48, and 72 h treatments, respectively. A dose-dependent increase of pro-apoptotic Bax protein in DU145 preincubated with coordination compound **58** was observed.

Majouga et al. reported mixed-valence Cu(II/I) copper compounds based on 2-thioimidazolones with promising antitumor activity [72] (Figure 17).

An MTT test on MCF-7 (breast cancer cell line of invasive breast ductal carcinoma), SiHa (human cervical cancer cells with the modal chromosome number of 71), and HEK 293 (human embryonic kidney cell line) cell lines showed promising antitumor ability of compounds **59**–**62** (Table 7). The ability of compound **60** to damage DNA was confirmed by tunnel assay. Compound **60** also proved to be an effective telomerase inhibitor. Nuclear accumulation of labeled coordination compound **63** was proven using fluorescent microscopy.

An in vivo investigation of antitumor activity was conducted using breast adenocarcinoma Ca-755 inoculated into mice lines C57BL/6 (female). Treatment began 48 h after vaccination with compound **60** (24 and 12 mg/kg/d) injected intraperitoneally at 24-h intervals for five days. Indicators of tumor growth inhibition for mice with a course of the test substance at a dose of 12 mg/kg was 46.1% on day seven after the end of treatment and 36.1% on day 14 after the end of treatment. For a dose of 24 mg/kg, it was 73.5% on day seven after the end of treatment and 59.5% on day 14 after the end of treatment, animal’s body weight loss did not exceed 10%. Telomerase inhibitor compound **60**, thus, proved to be an effective antitumor agent.

### 2.4. S/S-Donor Ligands

The cytotoxic activity of copper coordination compounds can occur both when a coordination compound solution is administered in vivo/in cell and when a nontoxic ligand is administered with the cytotoxic coordination compound forming in situ (this approach has already been described for elesclomol in Section 2.2). Another example of the formation of a coordination compound during therapy is disulfiram (DSF), an FDA-approved drug for treating alcoholism. In recent years, the drug has attracted much attention as an antitumor inducer of ROS formation acting in combination with copper gluconate [73]. Disulfiram alone has a negligible effect on tumor cells, but in the presence of Cu(II) ions in a nanomolar range, it is effective against a wide range of tumor cell lines, as shown in [74]. In vivo administration of folate-targeted nanoparticles with encapsulated DSF to animals with subcutaneous models of breast cancer led to a decrease in tumor growth [75]. It was repeatedly shown [76] that the cytotoxic effect of the DSF emerges as a result of in situ formation of the coordination compound Cu-DSF. But a thorough study of DSF metabolism led to the conclusion that the coordination compound Cu-DSF does not in fact exist [77]. It was found in [78] that even in an aqueous solution, DSF does not form a coordination compound with copper and in fact decomposition into diethyldithiocarbamate (DDC) occurs. The resulting diethyldithiocarbamate (DDC) reacts with Cu(II) to form copper diethyldithiocarbamate **64**, Cu(DDC)_2_ (Figure 18).

Dithiocarbamates are a known class of copper chelating agents that exhibit significant antitumor activity in vitro against various tumor cell lines [79]. We note that the use of DSF in combination with copper gluconate is in clinical trials as a breast cancer treatment scheme [80]. Liu et al. studied the mechanism of action in detail and reported that the disulfiram/copper mixture inhibits Bcl2 and induced Bax protein expression in all GBM cell lines, induces ROS activity, activates the apoptosis JNK pathway, causes ROS-dependent activation of the JNK and p38 pathways, and inhibits NFkB and ALDH activity [81]. In a comment to Liu et al., Cvek recalled successful trials for breast cancer using diethyldithiocarbamate in 1993 [82] and called for a return to undeservedly forgotten clinical trials of this inexpensive and effective drug [83]. Accordingly, Cvek, disulfiram could be used for treating brain tumors and even other cancers.

At present, DSF/Cu antitumor activity is still of interest. Duan et al. proposed synergistic breast tumor therapy via codelivery of doxorubicin and disulfiram cell killing using pH-sensitive core-shell-corona nanoparticles [84].

In vivo antitumor efficacy was proved using 4T1 tumor-bearing mice. The tumor inhibiting rate was 34.81% for the DSF-treated group, 68.27% for the DOX-treated group, 80.92% for the DSF + DOX-treated group, and 89.27% for the Co-NPs-treated group.

Wencheng Wu published a delicate pH-sensitive Cu/DSF delivery approach based on constructing mesoporous silica nanoparticles enriched with covalent-bonded Cu^2+^ ions and physically bonded DSF [85]. Under mild acidic conditions of the tumor microenvironment, a rapid biodegradation of nanoparticles was assumed to occur with subsequent Cu^2+^ and DSF release and an instantaneous chelation reaction leading to the formation of a cytotoxic agent. This approach was successfully tested, proving its efficacy in vitro and in vivo. Treating 4T1 tumor-bearing female BALB/C nude mice with 3.75 mg/kg of DSF dose (as part of the developed formulation) led to 71.4% tumor growth inhibition after two weeks of treatment, and no significant body-weight changes was observed.

Yiqiu Li et al. reported a DSF/Cu combination to induce anti-NPC activity through a joint action of multiple apoptosis pathways, such as an increasing chloride channel-3 protein expression, inducing ROS production, and decreasing NF-KB-p65 expression [86], and inhibiting the expression of α-SMA in cancer-associated fibroblasts (CAFs) [87]. In vivo, DSF/Cu combined with cisplatin (CDDP) therapy was well tolerated and could significantly suppress the growth of nasopharyngeal (NPC) tissues [88]. McMahon et al. recently summarized all drug-delivery approaches and formulations for DSF/Cu combinations [89]. Several active clinical trials testing the anticancer efficacy of DSF against various cancers are underway, such as germ cell tumor treatment [90] and breast neoplasm female and metastatic breast cancer [80,91]. Therefore, Disulfiram is a copper-based antitumor drug with great therapeutic potential for treating malignant tumors.

Zinc pyrithione is an agent with antimicrobial activity [92]. The ligand pyrithione itself has no antiproliferative activity, but copper coordination compound **65** based on it demonstrated antitumor activity [93] (Figure 19). A study of cytotoxicity of compound **65** on breast cancer cells (MCF-7) showed significant activity of the coordination compound with IC_50_ = 0.375 μM. Activity was also detected in U266 multiple myeloma cells with IC_50_ = 0.130 µM and in HepG2 liver cells with IC_50_ = 0.495 µM.

The cytotoxic effects of compound **65** were evaluated ex vivo on bone marrow cells from patients with acute myeloid leukemia (AML) and on mononuclear cells from peripheral blood of healthy volunteers. In AML patients, (CTR) CuPT and Bortezomib, respectively, reduced the viability of primary monocyte cells with an average IC_50_ of 57.03 and 20.50 nM, while in an experiment with healthy cells (CTR), the average IC_50_ was respectively estimated at 101.08 and 74.23 nM. A 12-h incubation of AML cells with coordination compound **65** in doses ranging from 0.25 to 0.75 μM led to apoptosis, which was shown by staining with annexin V/PI by flow cytometry. Coordination compound **65**, thus, showed efficacy in AML therapy as compared with clinically used Bortezomib.

### 2.5. N-, O-, and S-Donor Ligands

The antitumor activity of copper coordination compounds with thiosemicarbazone has been known since the 1960s. Zhang H et al. reported Cu(II) coordination compound **66** with thiosemicarbazide 8-hydroxyquinoline-2-carboxaldehyde Cu(HQTS) and **67** 8-hydroxyquinoline-2-carboxaldehyde-4,4-dimethyl-3-thiosemicarbazide Cu(HQDMTS) (Figure 20) [94]. An IC_50_ of 0.13 μM for compound **66** and 0.64 μM for compound **67** was obtained as a result of a study on SK-N-DZ neuroblastoma cell populations.

Hancock et al. reported simultaneous administration of a thiosemicarbazone copper-chelating agent in combination with a copper salt [95]. In situ formation of a cytotoxic coordination compound was assumed, as in the case of Disulfiram and elesclomol (Figure 21). A study of the cytotoxic effect of thiosemicarbazone ligand **L68** in combination with an equimolar amount of copper chloride showed the induction of a cytotoxic effect by oxidative stress, glutathione depletion, and ROS formation.

In vivo toxicological studies have shown that the administration of 100 mg/kg of ligand in 100% DMSO does not lead to a decrease in tumor mass in mice. An equimolar mixture of a ligand **L68** with copper chloride in mice showed the maximum tolerated dose of 15 mg/kg, and the administration of 3 mg/kg intravenously two times a day during five days resulted in a significant (42%) decrease in tumors (human leukemia cell line HL60), with less than 20% body weight loss. An ability to deplete glutathione in the cell by the same mechanism as arsenic oxide (As_2_O_3_) was also proven. The authors suggested that a combination of ligand **L68** with a copper salt can be used in conjunction with classical chemotherapeutic agents (cisplatin or Bortezomib).

Carcelli et al. reported synthesis and the cytotoxic activity of Cu(II) coordination compounds with variously substituted salicylaldehyde thiosemicarbazone ligands [96] (Figure 22). Inhibition doses in the low nanomolar range were found in some cases.

The in vitro activity of copper complexes on a pair of human colon cancer cell lines (LoVo/LoVo-OXP) showed anticancer activity of the coordination compounds including on the oxaliplatin-resistant human colon cancer cell line of supraclavicular lymph node metastasis cell line LoVo-OXP (Table 8).

Three-dimensional MTT tests on colorectal adenocarcinoma cell line HCT-116, human pancreatic adenocarcinoma cell line **PSN-1** spheroids also showed a high antiproliferative activity of coordination compounds **69**–**74** and a significant efficacy as compared with cisplatin (Table 9).

Cellular uptake and distribution were estimated using ICP-MS. A direct correlation between cellular accumulation and cytotoxic potency was not found by comparing uptake and cytotoxicity data in LoVo human colon cancer cells. Compounds **69**–**71** accumulated, substantially, in the mitochondria fraction and to a lesser extent in cytosolic fractions.

To estimate if compounds **69**–**74** cause DNA damage, DNA fragmentation was evaluated using alkaline single-cell gel electrophoresis (comet assay), and no DNA fragmentation was observed. In addition, no glutathione depletion was observed, and an ROS evaluation confirmed that TSC complexes did not provoke a substantial increase of cellular ROS levels. Therefore, the promising antitumor activity of these coordination compounds is not associated with the redox activity of copper ions. In clarification of the mechanism of cytotoxic action, the complexes were found to inhibit the protein disulfide isomerase (PDI) enzyme. Therefore, it was hypothesized that the coordination compounds **69**–**74** interfere with PDI activity, possibly inhibiting its disulfide bond catalytic activity.

The in vivo antitumor activity of compound **69** was evaluated in a solid tumor model, the highly aggressive syngeneic murine Lewis lung carcinoma (LLC). Tumor growth inhibition induced by compound **69** was compared with that of cisplatin (Table 10).

A 6 mg/kg dose of coordination compound **69** induced tumor growth inhibition of about 74%, similar to cisplatin dosed at 1.5 mg/kg, but the time course of changes in body weight indicated that cisplatin induced elevated anorexia. In contrast, treatment with compound **69** did not induce a substantial body weight loss (<10%) throughout the therapeutic experiment. Once again, these results confirm the higher biocompatibility of copper-containing anticancer drugs as compared with classical platinum therapy.

Kongot et al. developed Cu(II) coordination compound **75** based on a ligand obtained by the reaction between S-benzyldithiocarbamate and 2-hydroxy-5-(phenyldiazenyl)benzaldehyde [97] (Figure 23). The ability of compound **75** to bind to bovine serum albumin (BSA) was tested using its own fluorescence quenching method (titration of a protein solution with a solution of a coordination compound). The binding constant of coordination compound **75** calculated using the Benesi–Hildebrand equation was Ka = 0.94 × 10^4^ M^−1^, which indicates a significant binding energy. The authors believe that the most probable reason for this affinity is the hydrogen bond between the amino groups of the amino acid residues of the protein and the phenolic oxygen of the ligand.

The cytotoxic activity of coordination compound **75** and the corresponding ligand was studied on human cervical cancer cells HeLa. Significant cell death was achieved in the concentration range of 2–10 μM. The calculated IC_50_ values were 4.46 μM for coordination compound **75** and 5.34 μM for the corresponding ligand. The selectivity of the drugs obtained was evaluated by comparing the cytotoxicity of the obtained compounds on healthy human cells HEK-293, which showed a rather high cell death at a concentration of 10 μM. On the basis of the obtained data, CC50 = 6.31 μM for coordination compound **75** and 10.90 μM for the ligand were calculated. Thus, coordination compound **75** was found to be selective for healthy cells at a concentration of IC_50_ = 4.46 μM obtained on HeLa cell lines, which also confirmed the selectivity of the coordination compound with respect to tumor cells.

### 2.6. Phosphine-Donor Ligands

Cu(I) phosphine-based coordination compounds attract wide attention due to high cytotoxicity, antibacterial, and anti-inflammatory properties [98]. The use of a phosphine ligand prevents oxidation and hydrolysis reactions due to a strong copper–phosphine interaction [33], which allows stabilization of copper in a monovalent state, providing divers biological activity.

Khan et al. [99] reported nine copper Cu(I) complexes based on thiphenilphosphine and thiourea (Figure 24).

The synthesized compounds were utilized in different biological assays, which showed antibacterial, antifungal, antilieshmanial, antioxidant, and cytotoxic properties against brine shrimps. Compounds **79** and **81** proved to be the most active molecules against bacteria, fungi, and the lieshmanial pathogen, in addition to having an excellent antioxidant activity.

Tapanelli et al. [100] reported two water-soluble Cu(I) phosphonate complexes compounds **84** and **85**, which showed activity against Plasmodium early sporogonic (Figure 25).

Coordination compound **84** with a more hydrophilic and less bulky tris(hydroxymethyl)phosphane showed inhibition of plasmodia growth at an early stage of the disease up to 85 percent, while a sterically bulk coordination compound **85** acted two to three times weaker in different replicas. At the same time, when the dose was reduced from 100 μM, the therapeutic effect disappeared completely. For similar gold and silver compounds, inhibition of parasite growth in 80–85% occurred already at concentrations of 10 μM.

Mashat et al. [101] reported four Cu(I) phenanthroline-phosphine coordination compounds **86**–**89** (Figure 26).

Coordination compounds **86**–**89** showed DNA intercalation ability. Strong DNA binding is provided by triphenylphosphine ligands with a substituent at the 4-position of the phenyl ring capable of forming hydrogen bonds. Coordination compounds **86**–**89** showed IC_50_ values of 25–91 μM on the MCF-7 cell line. The highest cytotoxicity was observed for compounds **87** and **89**, which showed the strongest binding to DNA.

Komarnicka et al. [102] reported four novel Cu(I) complexes based on hydroxymethyldiphenylphosphine **90**–**93**, and four mixed sparfloxacin (HSf), i.e., hydroxymethyldiphenylphosphine coordination compounds **94**–**97** (Figure 27).

The cytotoxicity of the complexes synthesized and ligands (diimines, phosphines and phosphine oxides as potential decomposition products), starting compounds (CuNCS and CuI) was tested in vitro towards two cancer cell lines, i.e., mouse colon carcinoma (CT26) and human lung adenocarcinoma (A549). All tested complexes showed greater cytotoxicity than the corresponding ligands and copper iodide. In the case of the A549 line, dmp complexes compounds **90** and **91** ~25 mkM were twice as active as the bq complexes compounds **92** and **93** ~75 mkM. The activity against CT26 was slightly higher but was similar for both types of complexes.

It was shown that the penetration of complexes into cells proceeds quickly and an increase in the incubation time of cells with drugs from 4 h to 24 h does not lead to an increase in cytotoxicity. Sparfloxacin moiety introduction into the structure of the phosphine ligand led to a two-fold increase in toxicity on both cell lines.

## 3. Drug-Based Copper Coordination Compounds

The redox activity of copper cations along with their therapeutic efficacy, biogenicity, and ability to coordinate with various donor atoms opens up possibilities for synthesizing coordination compounds based on FDA-approved clinically used drugs with resultant target molecules having multiple biological effects. Copper coordination compounds based on ciprofloxacin [103], isoniazid [104], doxorubicin [105], indomethacin [106], and clioquinol [107] have been described. Coordination of copper cations with a drug molecule can change the pharmacodynamics of the drug and also enhance and complement therapeutic activity. A copper-containing gel based on indomethacin has shown increased activity as a local anti-inflammatory agent as compared with indomethacin [108]. A Cu(II) coordination compound based on indomethacin is capable of activating a copper-dependent opioid receptor and has a more effective analgesic effect than morphine with adjuvant arthritis after subcutaneous and oral administration [109].

Cu(II) coordination compound **98** with the anti-inflammatory drug Diclofenac was described by [110] (Figure 28). Diclofenac is one of the first anti-inflammatory NSAIDs used in medicine. Epidemiological studies have shown that chronic inflammation predisposes patients to the development of tumor diseases and that the long-term use of non-steroidal anti-inflammatory drugs reduced the risk of developing malignant neoplasms.

The cytotoxic effect of compound **98** was evaluated using the Uptiblue test on the following four human cell lines: human dermal fibroblast (HDF), human keratinocyte cell line HaCaT, and human colon adenocarcinoma cell lines SW620 and HT29. A comparison of the cytotoxic activity of the initial Cu(II) salt, Diclofenac, and coordination compound **98** ([Cu(diclofenac)_2_(H_2_O)_2_]) showed that the initial compounds have no cytotoxic activity (survival on tumor cell lines does not exceed 70% at a concentration 200 μM), while coordination compound **98** exhibits cytotoxicity on human colon adenocarcinoma cell lines SW620 and HT29 with the respective IC_50_ of 100 and 93 μM. Compound **98** ([Cu(diclofenac)_2_(H_2_O)_2_]) is also the first Diclofenac-based coordination compound synthesized in a 100% aqueous medium.

Fenoprofen is a non-steroidal anti-inflammatory drug, a propionic acid derivative with anti-inflammatory, analgesic, and antipyretic effects. Gumilar et al. reported Cu(II) coordination compounds based on fenoprofen, caffeine, and DMF as coligands [111] (Figure 29). The coordination compounds Cu_2_(Fen)_4_(caf)_2_ (Fen-fenoprofenate anion; caf-caffeine) **95** and Cu_2_(Fen)_2_(DMF)_2_
**96** have an analgesic effect, confirmed by studies in vitro and in vivo.

The analgetic properties of coordination compounds **99** and **100** were also of interest. The visceral analgesic action on acetic acid-induced pain of both complexes was five to seven times more potent than fenoprofen salt at the same fenoprofen dose (20 mg/kg). Moreover, both complexes showed longer onset and shorter duration of writhing than fenoprofen salt (data not shown). This indicates that compounds **99** and **100** present a strong analgesic activity for visceral pain.

Kovala-Demerzi et al. reported coordination compound **101** based on mefenamic acid ([Cu (Mef)_2_(H_2_O)]_2_) [112] (Figure 30). The cytotoxic activity of coordination compound **101** was tested in vitro on breast cancer cell line of invasive breast ductal carcinoma MCF-7, human bladder carcinoma cell line T24, lung cancer cell line of adenocarcinomic human alveolar basal epithelial cells A-549, and the mouse fibroblast cell line L-929. Coordination compound **101** exhibited greater cytotoxic activity as compared with the NSAID mefenamic acid (IC_50_ increased by two to six times). The IC_50_ values shown for **101** on the MCF-7 and L-929 tumor cell lines were compared with cisplatin (IC_50_ values were less than for cisplatin by 2.8 times for MCF-7 and 8.0 times for L-929). The coordination of mefenamic acid with copper (II) leads to the formation of an octahedral coordination compound, an increase in the cytotoxic activity of the initial ligand, and also new modes of cytotoxic action. Unfortunately, the low solubility of compound **101** prevents measuring anti-inflammatory effects in vivo.

Xiangchao Shi et al. developed copper (II) coordination compounds **102** and **103** based on a phenanthroline derivative and aspirin [113] (Figure 31). Compound **102** effectively induces mitochondrial dysfunction and promotes early apoptosis in ovarian cancer cells. It also inhibits the expression of cyclooxygenase-2 (COX-2), a key enzyme involved in the inflammatory response. A similar coordination compound **103** CuL without an aspirin ligand has a similar effect on cell redox homeostasis and cell cycle progression, but its cytotoxic activity is relatively low because its effect on mitochondrial function is mild and it cannot inhibit COX-2.

The IC_50_ value on ovarian cancer cell line of ovarian serous cystadenocarcinoma SKOV-3, cervical cancer cell line HeLa, and human normal kidney cell line HK-2 for coordination compound **102** is 10–30% less than that of compound **103**, which does not contain an aspirin fragment, and three to five times less than that of the ligand (Table 11). Coordination compound **102** was shown to have an antiproliferative effect through DNA degradation and mitochondrial dysfunction. The introduction of the aspirin moiety not only increases the antitumor efficacy of the drug but also reduces the inflammatory threat.

The COX-2 level in lipopolysaccharide-stimulated RAW macrophages was investigated on a flow cytometer after treatment with coordination compounds **102** and **103**, and the anti-inflammatory potential of coordination compound **102** was confirmed.

Nitroimidazole derivatives are widely used drugs with multiple pharmacological effects such as antifungal [114], antibacterial [115], and cytotoxic [116]. They also represent a class of hypoxia indicators that have been investigated for hypoxia-selective cytotoxicity and radiosensitization of hypoxic cells [117]. The effectiveness of these molecules depends on the generation of a nitroradical anion by intracellular reduction, which makes 5-nitroimidazoles suitable for penetration into cells by passive diffusion, creating a favorable concentration gradient. Once inside the cell, the nitroradical anion interacts with DNA and destroys the double helix. Cu(II) coordination compounds **104** and **105** and Cu(I) coordination compounds **106** and **107** with ligands based on 5-nitroimidazole were synthesized in [118] (Figure 32).

MTT tests on oxaliplatin-resistant and non-resistant human colon cancer cell line of supraclavicular lymph node metastasis LoVo-OXP and LoVo of compounds **104**–**107** showed a significant increase of antiproliferative activity of Cu(II) compounds **104** and **105** and Cu(I) compounds **106** and **107** as compared with ligands in monolayer cultures of various lines of human tumor cells. Water-soluble Cu(I) coordination compound **106** showed higher cytotoxicity as compared with Cu(II) coordination compound **104** (Table 12). The data obtained indicates that water-soluble Cu(I) coordination compounds have a better cellular accumulation than the Cu(II) analogues. This hypothesis was tested using AAS, and intracellular accumulation of coordination compound **106** (R = H) was shown to be better than the Cu(II) analogue.

Nitroimidazole derivatives are a promising platform for developing biologically active coordination compounds, and water-soluble Cu(I) coordination compounds capable of high cellular accumulation are a promising alternative to the classical Cu(II) coordination compounds, showing a significant improvement of the cytotoxic potency.

The 8-hydroxyquinoline (8-HQ) derivatives comprise a class of antifungal or antimicrobial agents. Developing Cu(II) coordination compounds with ligands based on oxyquinolones opens up opportunities for designing agents with multiple biological activity. Tardito et al. reported a halogenated clioquinol (CQ), which is an analogue of 8-HQ, and copper coordination compounds **108**–**115** based on it [119] (Figure 33). MTT data for coordination compounds **108**–**115** on cervical cancer cell line HeLa and human prostate cancer cell line PC3 are given in Table 13.

A structure–activity relationship (SAR) study was conducted. Cellular accumulation of the drug can occur via active transport, as suggested by the hCTR1 copper transporter for cisplatin [120], or by passive diffusion through the plasma membrane, in which case the drug should be endowed with appropriate lipophilicity to pass through the cell membrane and reach a sufficient intracellular concentration. An excessively lipophilic compound accumulates in the membrane, while greater hydrophilicity prevents interaction with the lipid bilayer. Of the studied derivatives, coordination compounds based on the most hydrophilic ligands **L108** and **L109** (5-SO3-8-HQ and 5-SO3-7-I-8-HQ) do not exhibit cytotoxic activity, while ligands of the most active coordination compounds are ligands with intermediate lipophilicity, namely, ligands **L110** to **L112** (8-HQ, 5,7-Me-8-HQ, and 5-Cl-8-HQ).

Coordination compounds **110** and **114** (Cu-CQ and Cu-8-HQ) were shown to inhibit the proteasome activity. MTT tests showed that coordination compound **114** (Cu-CQ) is at least 10 times more cytotoxic than ligand **L114** administered separately when tested for 48 h on HeLa cell lines (IC_50_ = 8.9 μM for coordination compound **114** and 93 μM for the ligand) and PC3 cell lines (IC_50_ = 9.0 μM for coordination compound **114** and >100 μM for the ligand).

Shah et al. demonstrated significant antibacterial activity of compound **114** (Cu-CQ) and its strong gain by copper ions [121]. The antibacterial activity of compound **110** was confirmed using Mtb-infected macrophages in the presence or absence of 7.5 μM CuSO_4_.

Copper coordination compounds based on 8-hydroxyquinolines exhibit both cytotoxic and antibacterial properties. The molecular design allows varying their lipophilicity and cellular accumulation. The results obtained together with the promising data obtained for elesclomol **L13** [38] confirm the promising use of copper coordination compounds in treating bacterial pneumonia caused by Mtb. The therapy certainly owes its success to the redox activity of copper cations. Both therapeutic regimens involving separate uses of copper and a ligand and in situ drug formation open up great opportunities for developing formulations, including those that are selective for healthy tissues.

Silva et al. reported a nanostructured lipid system for low-soluble isoniazid-based copper complexes compounds **116** ([CuCl_2_(INH)_2_]·H_2_O, **117** [Cu(NCS)_2_(INH)_2_]·5H_2_O, and **118** [Cu(NCO)_2_(INH)_2_]·4H_2_O with antimycobacterial activity (Figure 34). The nano-sized drug delivery systems increased their antimycobacterial activity, decreased cytotoxicity against the Vero cell line, and consequently improved the selectivity index [122].

A recent study on the cyto-genotoxicity of Cu(II) compounds **116**–**118** with INH was conducted by Fregonezi et al. and also concluding that the compounds show no cytotoxicity in therapeutic doses [123].

## 4. Natural Product-Based Copper Coordination Compounds

Natural products (NPs) have attracted lots of attention as biological active ligands for copper coordination compounds, due to the fact that nearly 60% of clinically approved anticancer drugs are associated with NPs. Advances of metal complexes with natural product-like compounds have been recently summarized by Heras et al. [124], herein a few examples of those design.

Fei et al. [125] reported two copper (II) complexes, compounds **119** and **120**, based on dehydroabietic acid (DHA), the main component of traditional Chinese medicine rosin (Figure 35).

The ability of compounds **119** and **120** to interact with calf thymus DNA (CT DNA) via intercalation, as well as albumin binding ability has been proven by various physicochemical methods. MTT assay illustrates that the selective cytotoxic activity of compound **119** was better than that of ligand **L119**, compound **120**, cisplatin, and oxaliplatin. The exposure of compound **119** to MCF-7 cells resulted in cell cycle arrest in G1 phase, apoptosis, mitochondrial dysfunction, and elevated ROS level, also compound **119** proved to induce apoptosis through intrinsic and extrinsic pathways, autophagy, and DNA damage in MCF-7 cells. Compound **119** is assumed to have the ability to resist metastasis and angiogenesis due to downregulation of VEGFR2, MMP-2, and MMP-9 expression levels in tumor cells.

Chen et al. [7] reported a coordination compound of copper **121** based on Hinokitiol, a natural monoterpenoid (Figure 36). A coordination compound was formed in situ while using equimolar mixtures of **L121** and CuSO_4_, as was previously described for Disulfiram, elesclomol and thiosemicarbazone ligand **L68**.

Ligand **L121** in the presence of CuSO_4_ induces striking accumulation of ubiquitinated proteins in A549 and K562 cells, which means that it is capable of inhibiting the activity of the 19S proteasomal DUBs much more effectively than it does the chymotrypsin-like activity of the 20S proteasome. Coordination compound **121** effectively induces caspase-independent and paraptosis-like cell death in A549 and K562 cells, and the resulted cell death has been proven to depend on ATF4-assosiated ER stress but not ROS generation.

## 5. Conclusions

Copper-containing coordination compounds are a promising class of drugs with multiple therapeutic effects from antitumor to anti-inflammatory activity. This review summarizes the successful use of copper coordination compounds as antitumor, antimalarial, antituberculosis, antifungal, and anti-inflammatory drugs.

An antitumor activity of copper coordination compounds was repeatedly proven in vivo. Imidazolin-4-one based coordination compound **60** showed 73.5% of breast adenocarcinoma Ca-755 growth inhibition in seven days of treatment with a 24 mg/kg dose; animal’s body weight loss did not exceed 10%. Thiosemicarbazone-based ligand **L68** + **CuCl_2_** showed 42% of monocytic leukemia HL60 growth inhibition in five days of treatment with a 3 mg/kg dose, with less than 20% body weight loss. The DSF/Cu nanoparticle delivery system showed 71.4% of breast cancer 4T1 growth inhibition after two weeks of treatment with a 3.75 mg/kg of DSF dose, and there were no significant body-weight changes observed. Thiosemicarbazone-based coordination compound **69** showed 74% of Lewis lung carcinoma (LLC) tumor growth inhibition in seven days of treatment with a 6 mg/kg dose, weight loss body weight loss did not exceed 10%.

Regarding the mechanism of antitumor activity, the vast majority of coordination compounds act through the ROS formation (**1**, **2**, **L13** + **Cu**, **42**, **43**, **45**–**47**, Disulfiram-based coordination compound **64**, **L68** + **Cu**, **119**), glutation depletion (**45**–**47**, **L68** + **Cu**, **69**–**74**), proteasome inhibition (**7**, **8**, **110**, **114**, **L121** + **Cu**), DNA degradation (**1**, **2**, **11**, **60**, **102**, **119**), DNA intercalation (**1**, **2**, **3**–**5**, **L13** + **Cu, 86**–**89**, **119**, **120**), apoptosis induction (**1**, **2**, **65**, **102**, **119**), and cell cycle arrest (**51**–**57**, **103**, **119**). Despite the fact that most coordination compounds of copper exhibit ROS-mediated cytotoxicity, examples of coordination compounds acting by other mechanisms are also described. Thus, the cytotoxic activity of coordination compounds **42**, **43**, **69**–**74**, **L121** + **Cu** is not associated with ROS formation and does not decrease under the influence of ROS inhibitors.

Redox-active drugs proved to be an effective supplement in addition to antituberculous drugs, or even being an independent therapy. Thus, elesclomol-based copper coordination compound **13**, Cu-CQ–based coordination compound **114**, and isoniazid-based coordination compounds **116**–**118** showed promising activity against *Mycobacterium tuberculosis.* Phosphine-based coordination compounds **79** and **81** showed a promising activity against bacteria, fungi, and the lieshmanial pathogen. In addition, 1,10-phenanthroline–based coordination compounds **32**–**39** showed higher antibiofilm activity than the clinically used Vancomycin, which is also associated with redox activity of copper cations.

The ability of copper coordination compounds to penetrate through the blood-brain barrier along with their stability in the bloodstream allows development of ^64^Cu-marked PET-imagine agents. Thiosemicarbazone-based coordination compounds **23**–**30** are Aβ-targeted PET-visualizers of Alzheimer’s disease showed the ability to rapidly crossing the blood-brain barrier, as well as good brain uptake and Aβ plaques affinity. In addition, CuATSM coordination compounds are hypoxia-sensitive PET-visualizers of malignant neoplasms, including head and neck cancer. Cu(II/I) redox potential was repeatedly proven to govern the hypoxia selectivity of CuATSM coordination compounds.

Anti-inflammatory properties of coper-containing coordination compounds are interesting due to the possibility of twin antitumor/anti-inflammatory drug development. Thus, aspirin-based coordination compound **102** showed COX-2 inhibition due to aspirin moiety, whereas coordination compounds **45**–**47** showed analgesic properties themselves.

It is also worth noting that using both a ligand and a copper salt is as effective as using a coordination compound. Copper-containing coordination compounds of disulfiram metabolite are always formed in situ, and the same approach has been successfully applied in vitro and in vivo to a number of compounds, such as **L13** + **Cu**, **L68** + **Cu**, **L108**–**L113** + **Cu**, and **L121** + **Cu**.

The redox activity of copper ions along with the their biogenicity, the stability of copper coordination compounds in the bloodstream, and the highly promising therapeutic results in vitro and in vivo prove the potential of copper coordination compounds to become widely used in clinical practice.

## Figures and Tables

**Figure 1 ijms-21-03965-f001:**
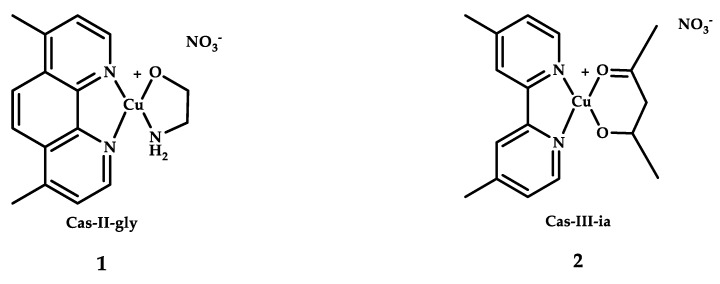
Chemical structures of Casiopeina II-gly **1** and Casiopeina-III-ia **2**.

**Figure 2 ijms-21-03965-f002:**
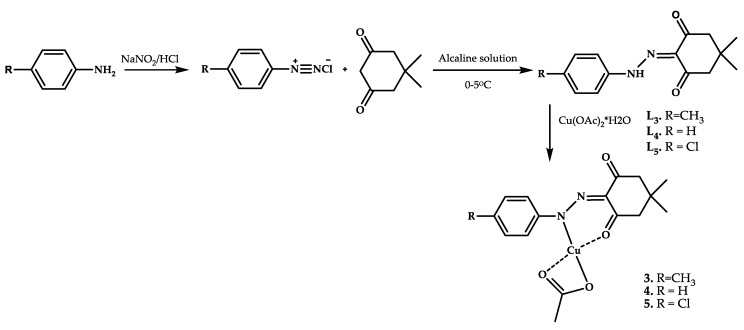
Ligand synthesis scheme and chemical structures of coordination compounds **3**–**5**.

**Figure 3 ijms-21-03965-f003:**
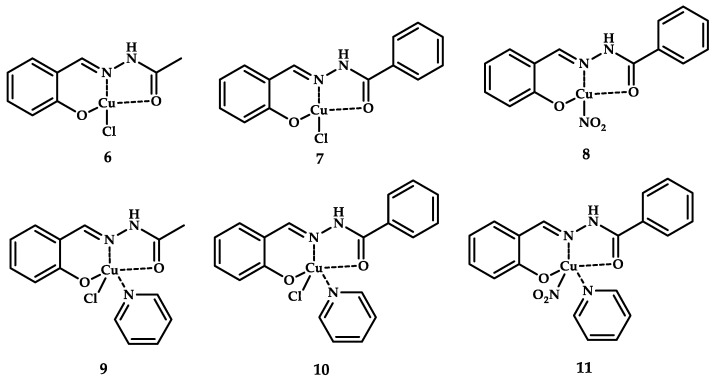
Chemical structures of coordination compounds **6**–**8** with Schiff-base ligands and **9**–**11** with a Py coligand.

**Figure 4 ijms-21-03965-f004:**
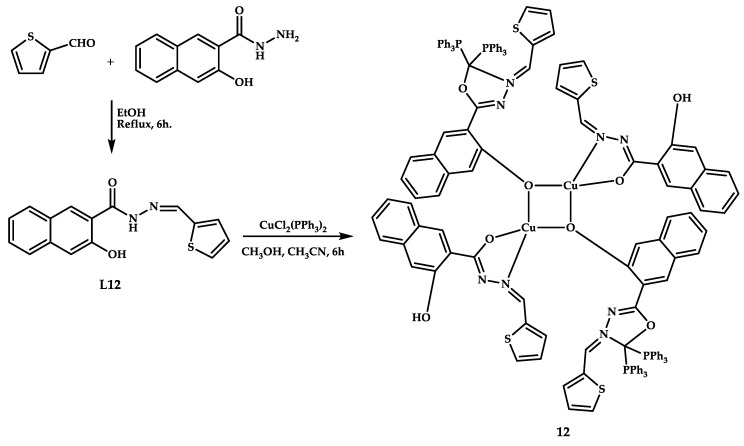
Synthesis scheme and chemical structure of coordination compound **12**.

**Figure 5 ijms-21-03965-f005:**
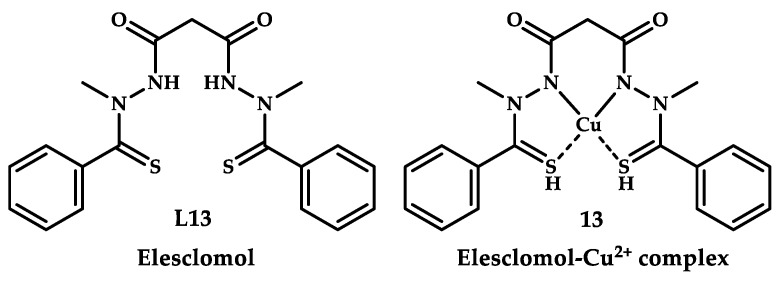
Chemical structure of elesclomol ligand **L13** and copper coordination compound **13**.

**Figure 6 ijms-21-03965-f006:**
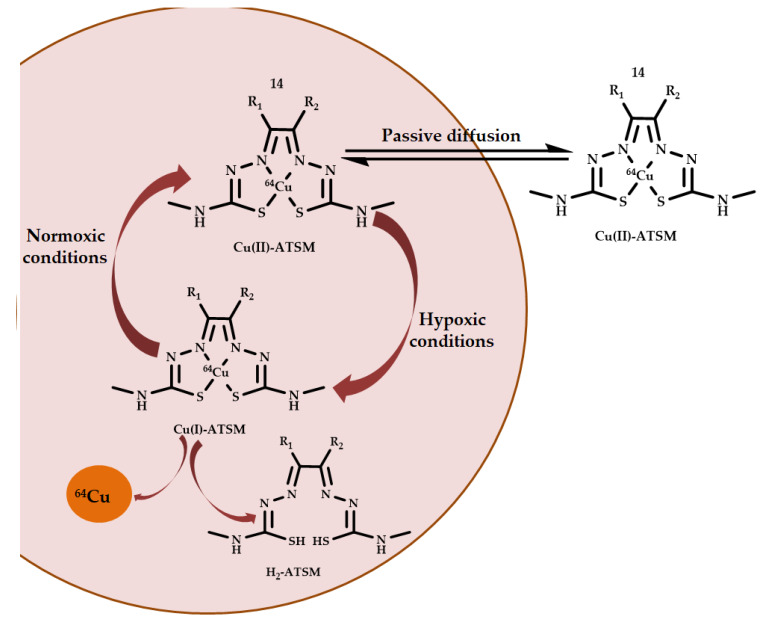
Chemical structure and cellular accumulation scheme of Cu-ATSM **14**.

**Figure 7 ijms-21-03965-f007:**
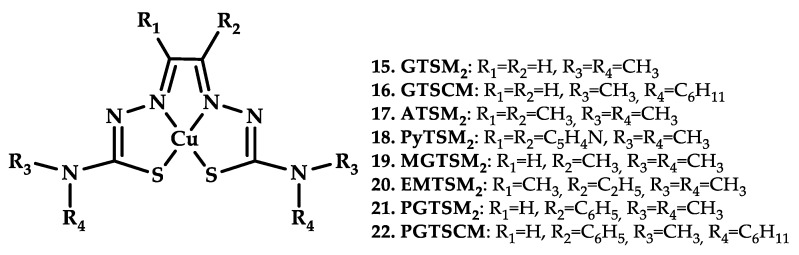
Chemical structures of bis(thiosemicarbazone)-based copper complexes **15**–**22**.

**Figure 8 ijms-21-03965-f008:**
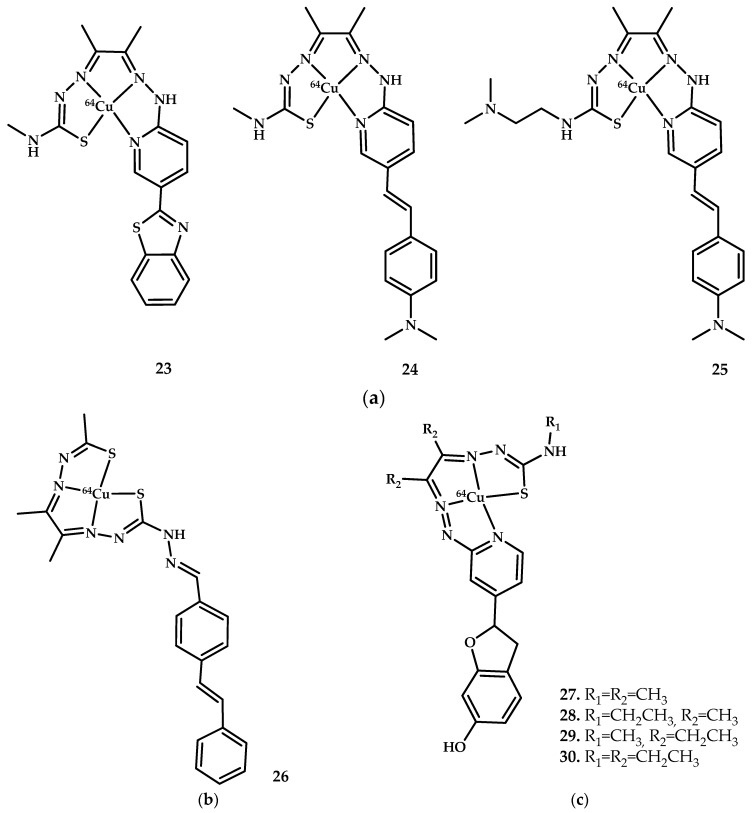
(**a**) Developed by Hickey et al. [60]. (**b**) Developed by SinChun Lim et al. [61]. (**c**) Developed by McInnes et al. [62]. Chemical structures of Cu-containing coordination compounds for PET imaging of Aβ plaques.

**Figure 9 ijms-21-03965-f009:**
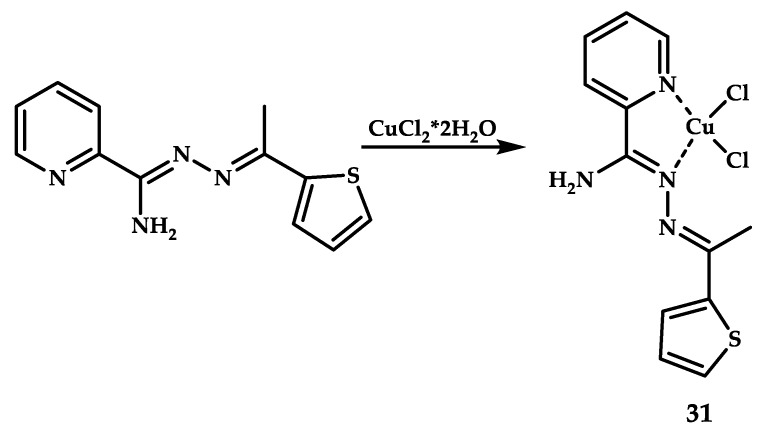
Synthesis scheme and chemical structure of coordination compound **31**.

**Figure 10 ijms-21-03965-f010:**
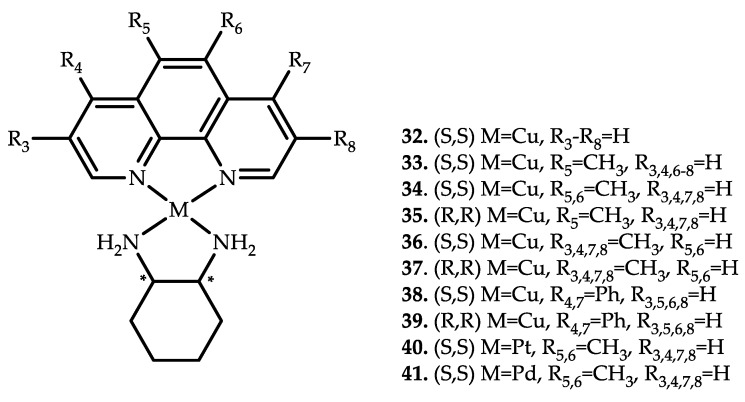
Chemical structure of coordination compounds **32**–**41**.

**Figure 11 ijms-21-03965-f011:**
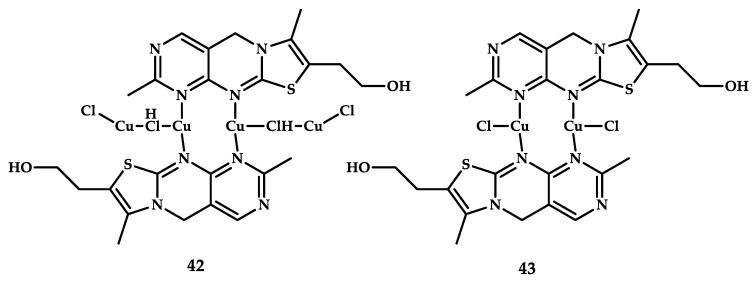
Chemical structure of coordination compounds **42** and **43**.

**Figure 12 ijms-21-03965-f012:**
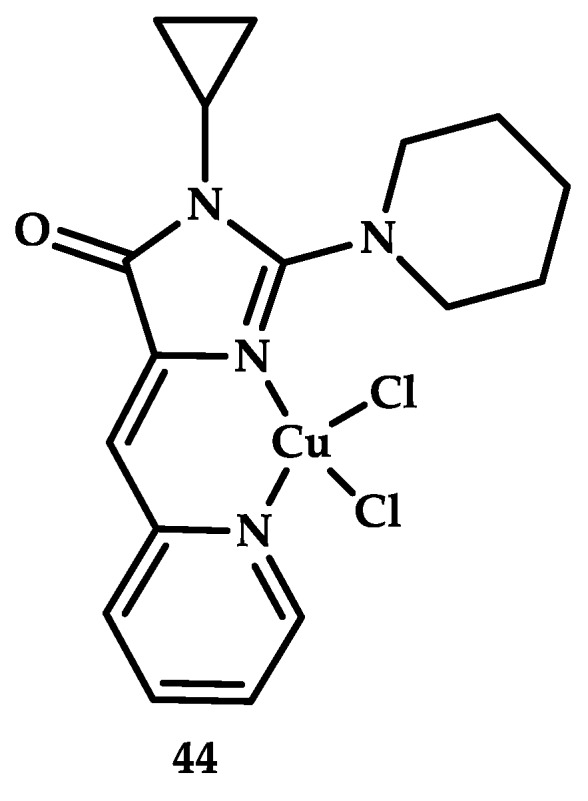
Chemical structure of coordination compound **44**.

**Figure 13 ijms-21-03965-f013:**
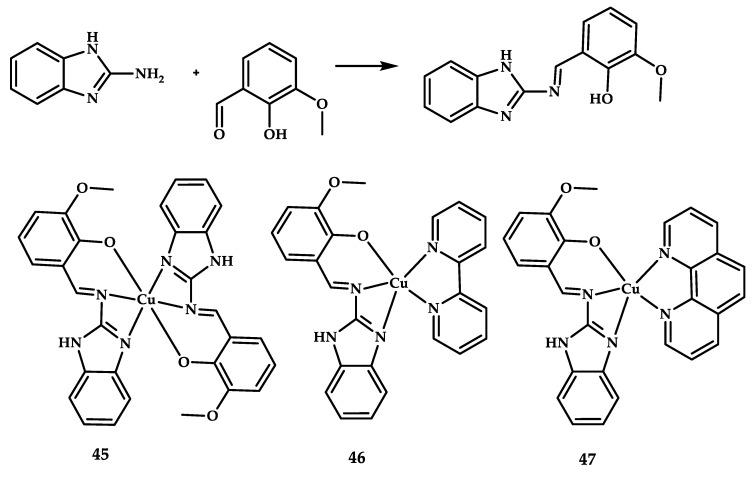
Ligand synthesis scheme and chemical structure of coordination compounds **45**–**47**.

**Figure 14 ijms-21-03965-f014:**
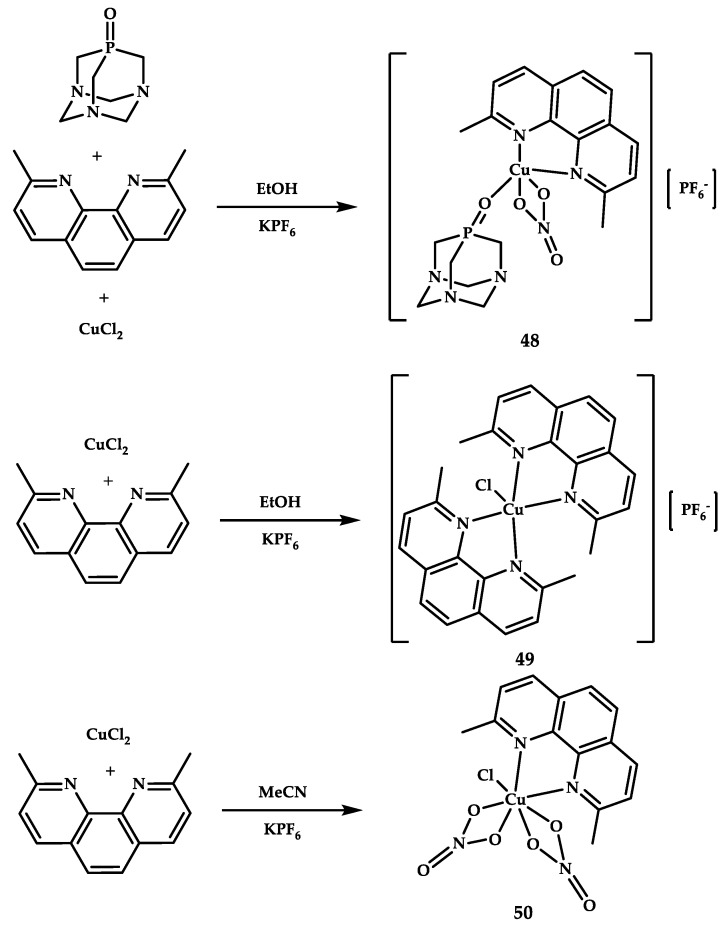
Synthesis scheme and chemical structure of coordination compounds **48**–**50**.

**Figure 15 ijms-21-03965-f015:**
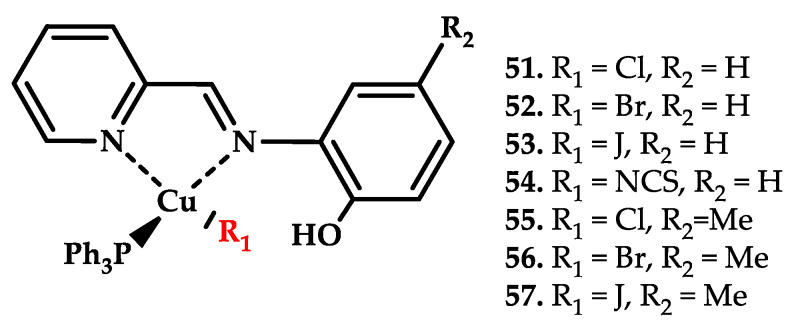
Chemical structure of coordination compounds **51**–**57**.

**Figure 16 ijms-21-03965-f016:**
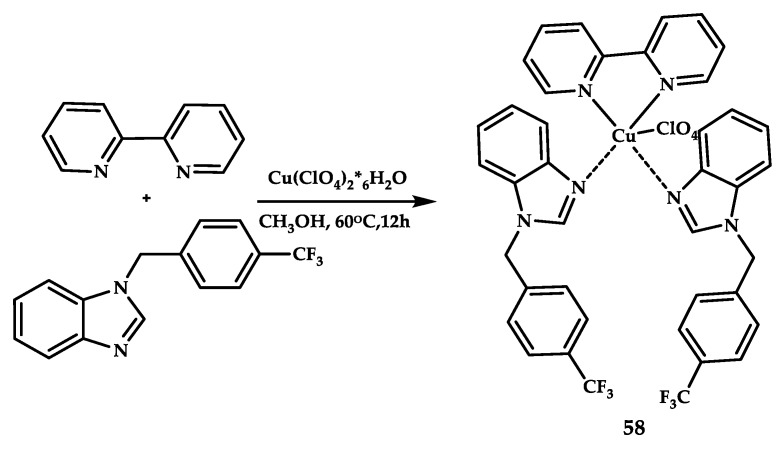
Synthesis scheme, chemical structure of coordination compound **58**.

**Figure 17 ijms-21-03965-f017:**
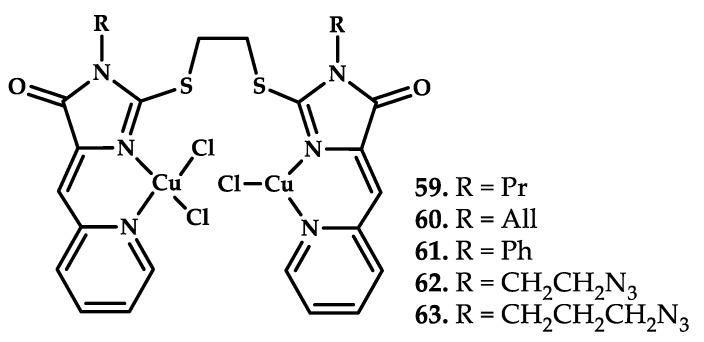
Chemical structure of coordination compounds **59**–**63**.

**Figure 18 ijms-21-03965-f018:**
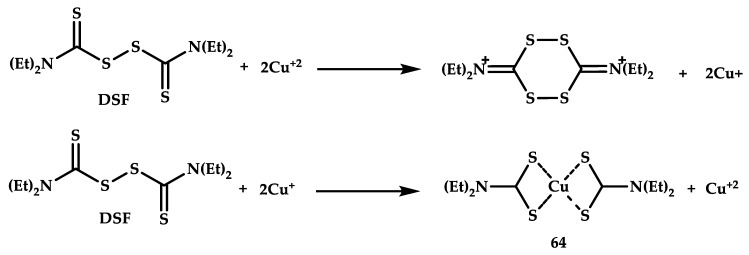
Schemes of the interaction of copper (II) with disulfiram (DSF) and the formation of a coordination compound **64**.

**Figure 19 ijms-21-03965-f019:**
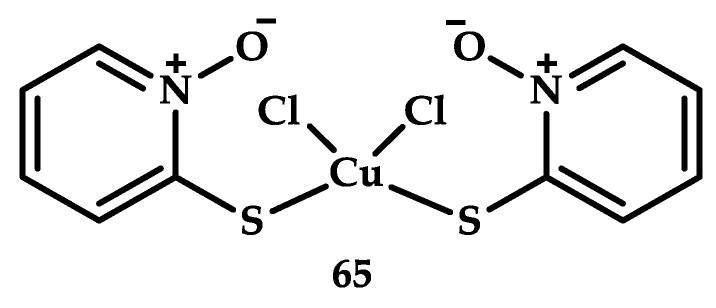
Chemical structure of the coordination compound **65** (CuPT).

**Figure 20 ijms-21-03965-f020:**
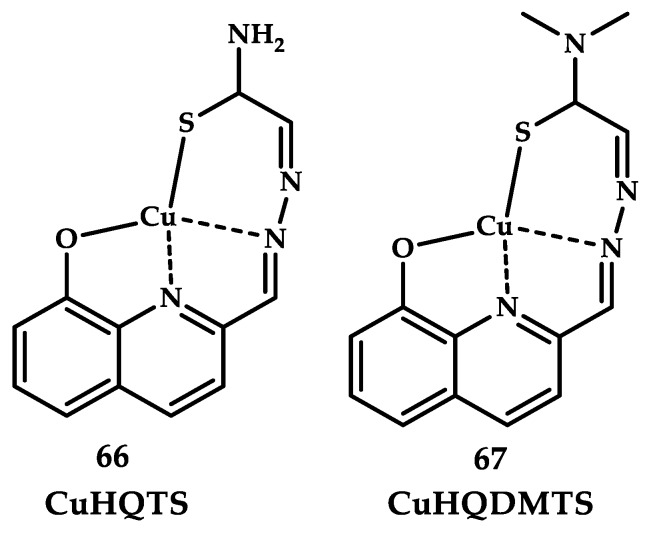
Chemical structures of the coordination compounds **66** and **67**.

**Figure 21 ijms-21-03965-f021:**
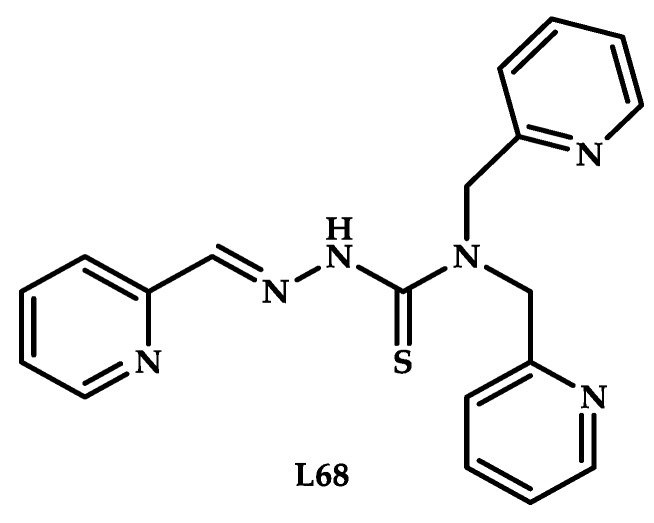
Chemical structure of organic ligand **L68**.

**Figure 22 ijms-21-03965-f022:**
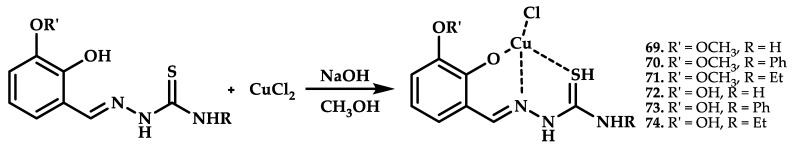
Synthesis scheme and chemical structure of coordination compounds **69**–**74**.

**Figure 23 ijms-21-03965-f023:**
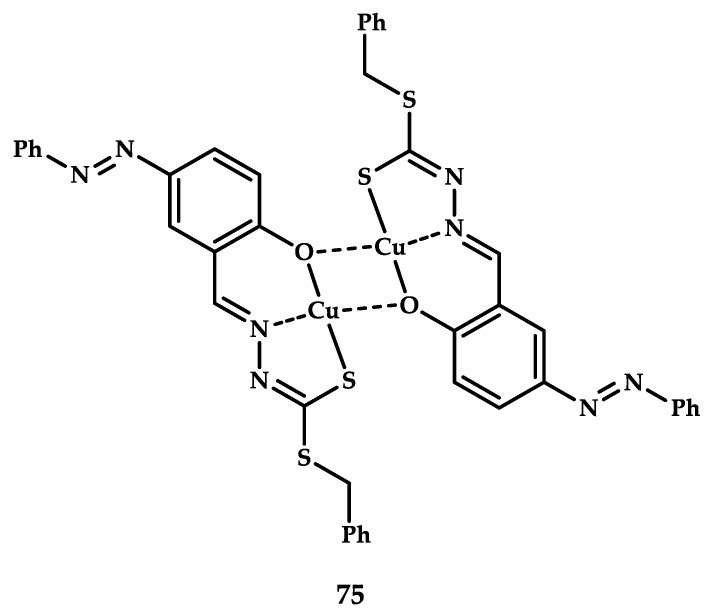
Chemical structure of coordination compound **75**.

**Figure 24 ijms-21-03965-f024:**
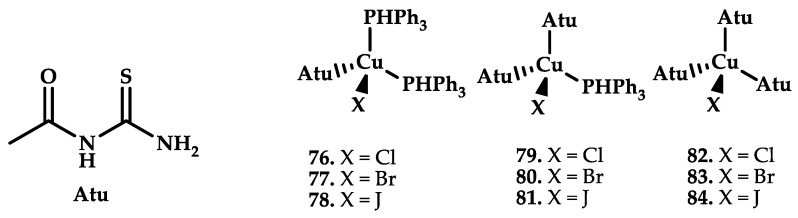
Chemical structure of coordination compounds **76**–**84**.

**Figure 25 ijms-21-03965-f025:**
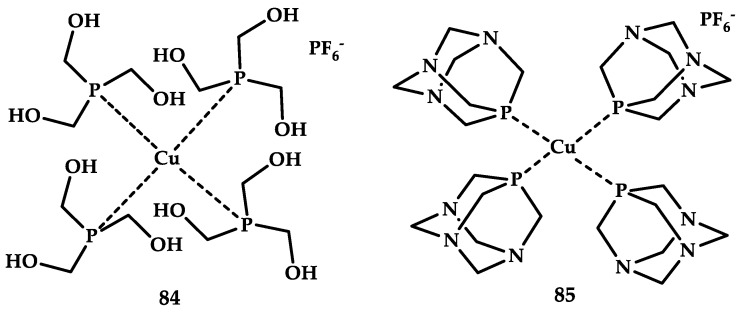
Chemical structure of coordination compounds **84** and **85**.

**Figure 26 ijms-21-03965-f026:**
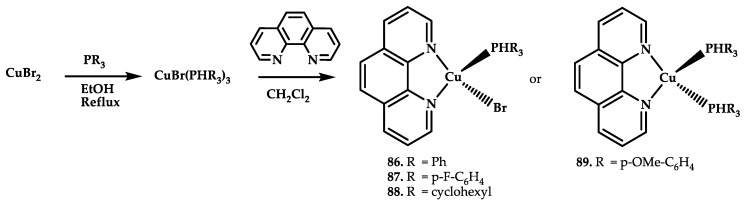
Chemical structure of coordination compounds **86**–**89**.

**Figure 27 ijms-21-03965-f027:**
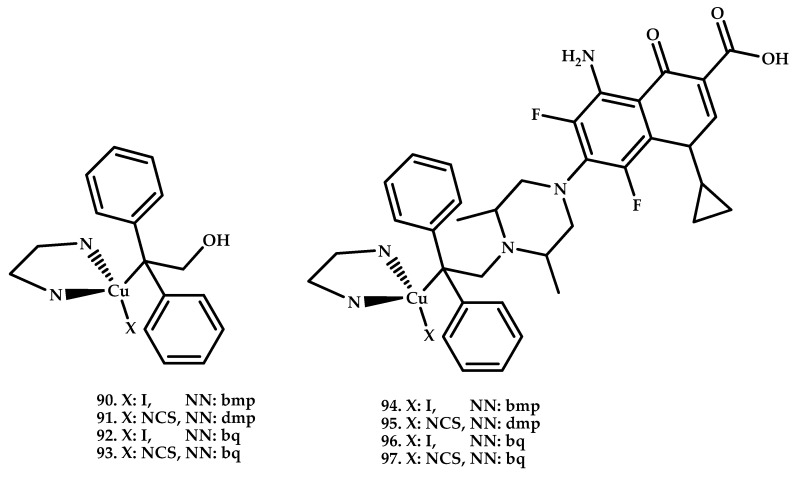
Chemical structures of Cu(I) coordination compounds based on hydroxymethyldiphenylphosphine **90**–**93**, sparfloxacin (HSf) hydroxymethyldiphenylphosphine **94**–**97**.

**Figure 28 ijms-21-03965-f028:**
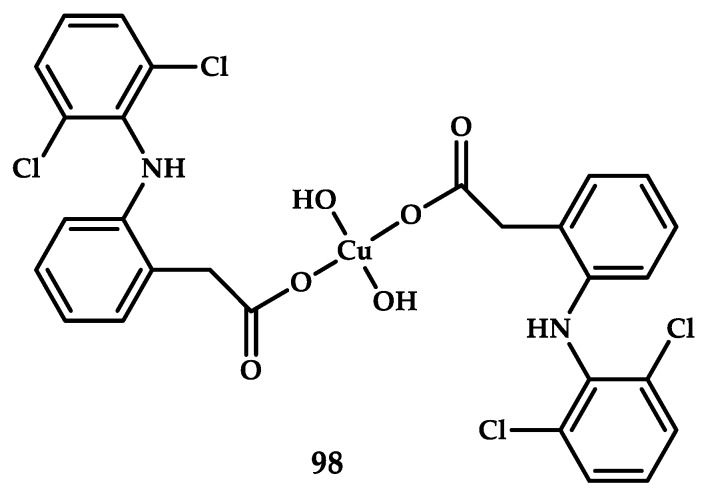
Chemical structure of coordination compound **98** [Cu(diclofenac)_2_(H_2_O)_2_] based on the NSAID Diclofenac.

**Figure 29 ijms-21-03965-f029:**
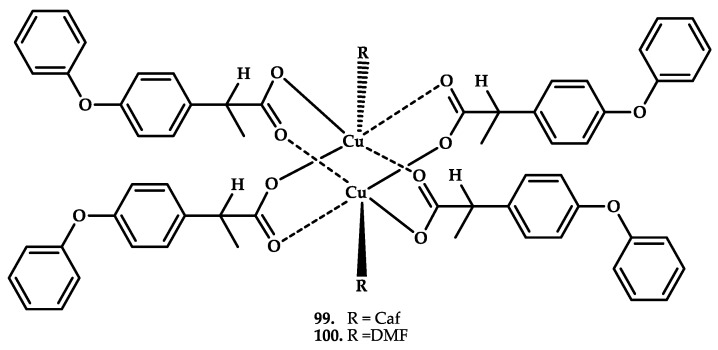
Chemical structure of coordination compounds **99** and **100** based on the NSAID drug fenoprofen.

**Figure 30 ijms-21-03965-f030:**
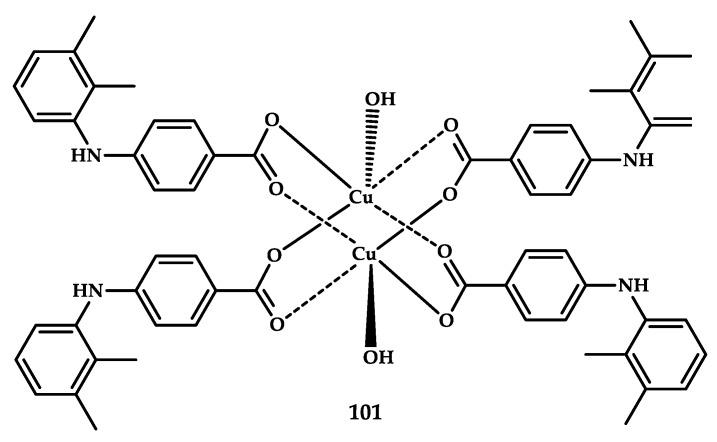
Chemical structure of coordination compound **101** based on the NSAID mefenamic acid.

**Figure 31 ijms-21-03965-f031:**
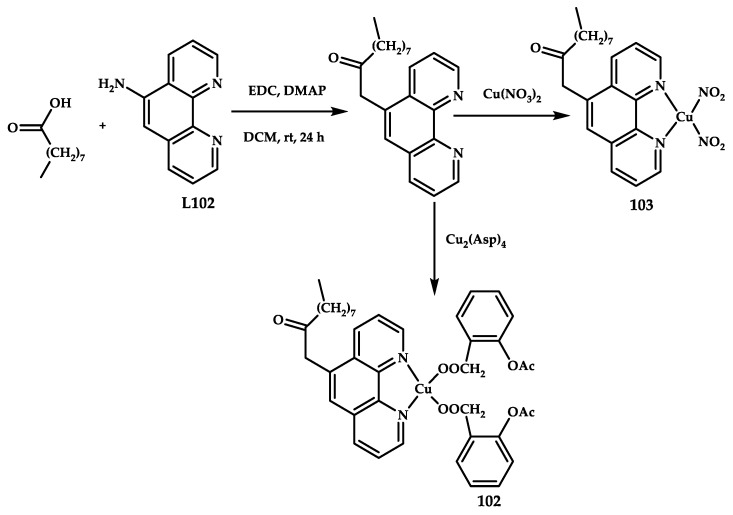
Synthesis scheme of copper coordination compounds **102** and **103** with mixed phenanthroline and the NSAID aspirin.

**Figure 32 ijms-21-03965-f032:**
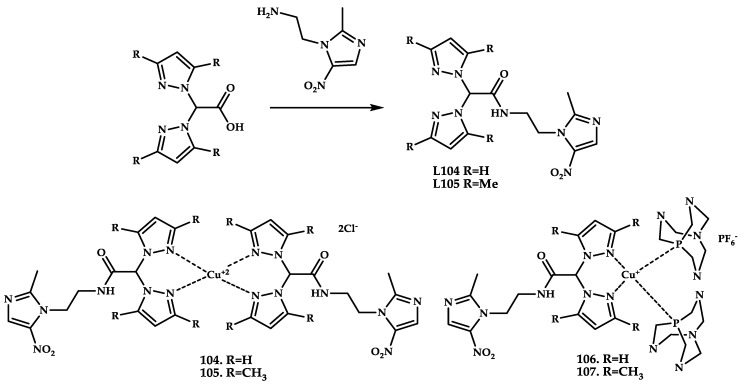
**L104** and **L105** ligand synthesis scheme and the chemical structures of coordination compounds **104**–**107** based on the antifungal drug Metronidazole.

**Figure 33 ijms-21-03965-f033:**
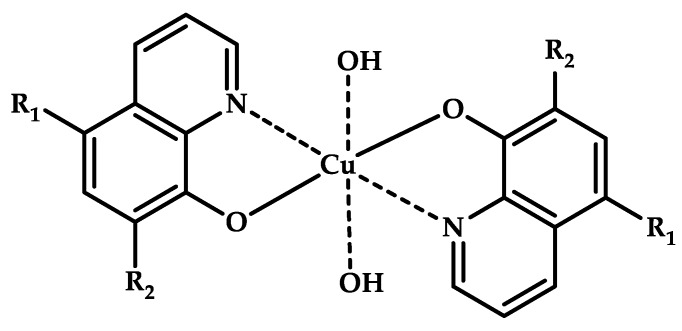
Estimated chemical structures of coordination compounds **108**–**115**.

**Figure 34 ijms-21-03965-f034:**
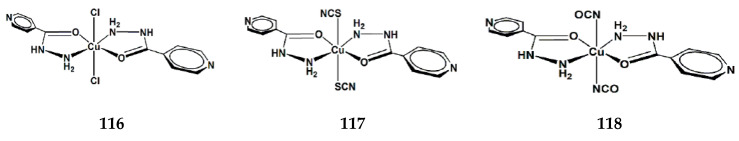
Chemical structures of coordination compounds **116**–**118** based on the antituberculosis drug isoniazid.

**Figure 35 ijms-21-03965-f035:**
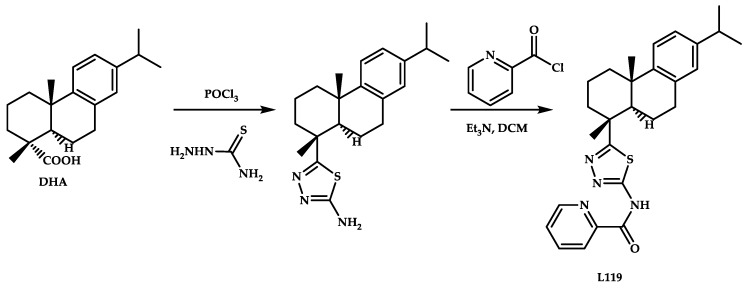
Synthesis scheme and chemical structure of coordination compounds **119** and **120** based on the dehydroabietic acid.

**Figure 36 ijms-21-03965-f036:**
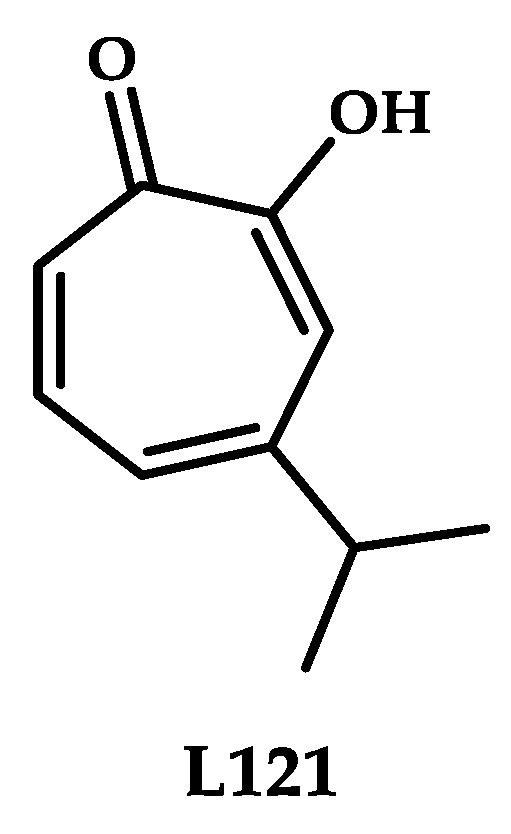
Chemical structure of Hinokitiol **L121**.

**Table 1 ijms-21-03965-t001:** MTT data of coordination compounds **3**–**5** and ligands **L3**–**L5** after 72 h of incubation [39].

IC_50_, µM ± S.D.					
Compound	HepG-2	MCF-7	Compound	HepG-2	MCF-7
**L3**	6.88 ± 0.5	27.19 ± 2.3	**3**	41.77 ± 2.7	26.57 ± 1.9
**L4**	7.60 ± 0.9	14.65 ± 1.5	**4**	11.80 ± 1.3	9.38 ± 1.0
**L5**	58.10 ± 3.4	63.13 ± 3.6	**5**	67.66 ± 3.8	46.75 ± 3.1

**Table 2 ijms-21-03965-t002:** Antibacterial and antifungal activities data of ligands **L3**–**L5** and Cu(II) coordination compounds **3**–**5** [39].

	*E. Coli*		*S. Aureus*		*C. Albicans*	
Compound	Diameter of Inhibition Zone (Mm)	% Activity Index	Diameter of Inhibition Zone (Mm)	% Activity Index	Diameter of Inhibition Zone (Mm)	% Activity Index
**L3**	13	52.0	18	78.3	21	80.8
**L4**	8	32.0	11	47.8	16	61.5
**L5**	6	24.0	5	21.7	8	30.8
**3**	3	12.0	8	34.8	14	53.8
**4**	9	36.0	16	69.6	19	73.1
**5**	NA	----	2	8.7	10	38.5
**Ampicillin**	25	100	23	100	NA	----
**Cloitrimazole**	NA	----	NA	----	26	100

**Table 3 ijms-21-03965-t003:** Effect of copper on antimycobacterial activity of ligand **L13** against Mtb H37Rv [48].

Medium Used	MIC (mg/L)
Middlebrook 7H9 *	4
Middlebrooks 7H12 *	4
HdB without CuSO_4_	>32
HdB (CuSO_4_ at 2 mg/L)	0.5

* amount of copper in medium was approximately 1 mg/L.

**Table 4 ijms-21-03965-t004:** Minimal inhibitory concentrations and hemolitic activity of coordination compounds **32**–**41** [64].

Compound	*S. Aureus*	*S. Aureus*	*E. Jaecalis*	*E. Coli*	*P. Aeruginosa*	*%* Lysis Rbcs
	MRSA252	MSSA209	NCTC775	NCTC86	ATCC27853	+/− (SD)
**32**	32	32	32	64	>128	2.0 (0.4)
**33**	32	32	8	64	>128	2.1 (0.1)
**34**	88	16	4	32	>128	2.6 (0.3)
**35**	8	4	2	32	>128	2.2 (0.7)
**36**	4	4	4	16	>128	2.5 (0.3)
**37**	4	4	4	16	>128	2.0 (0.3)
**38**	2	2	2	16	>128	3.1 (0.2)
**39**	2	2	2	16	>128	ND
**40**	128	32	4	16	>128	ND
**41**	64	64	16	32	>128	ND
**Vancomycin**	0.25	0.5	0.5	ND	ND	2.6(0.2)
**Chloramphenicol**	IG	16	4	2	128	ND
**CuCl_2_*2H_2_O**	>128	>128	>128	>128	>128	2.0 (0.3)

**Table 5 ijms-21-03965-t005:** MTT data of coordination compounds **48**–**50** after 72 h of incubation [69].

IC_50_, µM ± S.D.							
Cell Line/Compound	48	49	50	Cu(NO_3_)_2_	PTA = O	Dmphen	CDDP
**NHDF**	0.57 ± 0.08	0.23 ± 0.03	1.72 ± 0.25	310 ± 47	Nd	nd	16.6 ± 2.1
**A549**	0.29 ± 0.01	0.28 ± 0.04	0.43 ± 0.06	155 ± 23	Nd	nd	33.3 ± 4.2
**HeLa**	1.12 ± 0.16	1.13 ± 0.17	0.43 ± 0.06	19.1 ± 2.9	Nd	720 ± 108	16.6 ± 3.1
**MCF-7**	0.57 ± 0.08	0.57 ± 0.08	3.45 ± 0.51	155 ± 23	Nd	nd	33.3 ± 4.2
**LoVo**	0.57 ± 0.08	1.13 ± 0.17	1.72 ± 0.25	38.8 ± 5.8	Nd	360 ± 54	9.12 ± 0.005

**Table 6 ijms-21-03965-t006:** MTT data of coordination compounds **51**–**57** on U87 cells after 72 h of incubation [70].

IC_50_, µM ± S.D.							
Compound	51	52	53	54	55	56	57
	32.7 ± 0.6	31.3 ± 1.7	20 ± 1.5	46.7 ± 0.4	37.6 ± 1.1	34.5 ± 0.9	25.4 ± 0.5

**Table 7 ijms-21-03965-t007:** MTT data of coordination compounds **59**–**62** after 72 h of incubation [72].

IC_50_, µM ± S.D.			
Compound	MCF-7	SiHa	HEK293
**59**	3.7 ± 1.6	3.0 ± 0.2	2.5 ± 0.4
**60**	2.1 ± 0.8	2.2 ± 0.7	2.3 ± 0.9
**61**	7.4 ± 1.4	3.9 ± 2.3	25.3 ± 1.2
**62**	13.4 ± 3.8	8.5 ± 0.4	12.7 ± 3.7
**Dox**	2.1 ± 0.8	2.0 ± 0.8	1.1 ± 0.1
**CDDP**	64.1 ± 3.9	-	12.4 ± 3.9

**Table 8 ijms-21-03965-t008:** MTT data of coordination compounds **69**–**74** after 72 h of incubation [96].

IC_50_, µM ± S.D.		
Compound	LoVo	LoVo-OXP
**69**	0.031 ± 0.001	0.004 ± 0.001
**70**	0.029 ± 0.008	0.030 ± 0.010
**71**	0.036 ± 0.009	0.008 ± 0.002
**72**	0.020 ± 0.001	0.020 ± 0.001
**73**	0.21 ± 0.08	0.09 ± 0.01
**74**	0.030 ± 0.001	0.02 ± 0.01
**Oxaliplatin**	2.17 ± 1.37	13.92 ± 1.68

**Table 9 ijms-21-03965-t009:** MTT on three-dimensional (3D) spheroids of HCT-15 and PSN1 cancer cell spheroids of the coordination compounds **69**–**74** [96].

IC_50_, µM ± S.D.		
Compound	HCT-116	PSN-1
**69**	1.08 ± 0.38	0.90 ± 0.02
**70**	3.56 ± 1.67	1.17 ± 0.11
**71**	1.25 ± 0.98	0.90 ± 0.30
**72**	1.17 ± 0.62	0.94 ± 0.27
**73**	1.69 ± 0.45	1.18 ± 0.23
**74**	1.28 ± 0.62	0.91 ± 0.01
**CDDP**	68.20 ± 4.57	52.60 ± 3.78

**Table 10 ijms-21-03965-t010:** In vivo antitumor activity of coordination compound **69** (cisplatin as a control) on Lewis lung carcinoma (LLC) tumor-bearing mice [96].

	Daily Dose (Mg/Kg)	Average Tumor Weight (Mean ± S.D., G)	Inhibition of Tumor Growth (%)
**Control**	**-**	0.459 ± 0.130	-
**69**	3	0.239 ± 0.080	48.0
**69**	6	0.118 ± 0.090	74.3
**CDDP**	1.5	0.114 ± 0.080	75.2

**Table 11 ijms-21-03965-t011:** MTT data of coordination compounds **102** and **103**, and the phenanthroline ligand after 72 h of incubation [113].

IC_50_, µM ± S.D.			
Compound	SKOV-3	HeLa	HK-2
**102** (with aspirin)	1.1 ± 0.6	1.5 ± 0.5	4.4 ± 0.5
**103** (without aspirin)	1.5 ± 0.4	1.8 ± 0.5	4.6 ± 0.8
**L102**	5.4 ± 1.2	6.8 ± 1.2	12.3 ± 1.6

**Table 12 ijms-21-03965-t012:** MTT data of coordination compounds **104**–**107** after 72 h of incubation [118].

IC_50_, µM ± S.D.		
Compound	LoVo	LoVo-OXP
**104** Cu(II)	5.9 ± 0.6	5.1 ± 0.5
**105** Cu(II)	4.3 ± 0.5	4.6 ± 1.0
**106** Cu(I)	2.1 ± 1.1	1.9 ± 0.9
**107** Cu(I)	4.9 ± 1.0	4.6 ± 0.8

**Table 13 ijms-21-03965-t013:** The structures of coordination compounds **108**–**115** based on 8-HQ and their cytotoxic activity data [119].

			IC_50_, µM ± S.D.	
Compound	Chemical Structure	LogP (Ligand)	HeLa	PC_3_
**Cu(5-SO_3_-8-HQ)** **108**	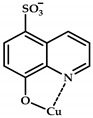	−0.21	n.d.	n.d.
**Cu(5-SO_3_-7-I-8-HQ)** **109**	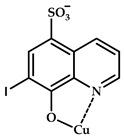	0.70	n.d.	n.d.
**Cu(8-HQ)** **110**	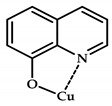	1.84	1.9	1.3
**Cu(5-Cl-8-HQ)** **111**	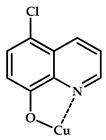	2.58	3.1	2.3
**Cu(5,7-Me-8-HQ)** **112**	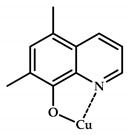	2.66	2.7	1.9
**Cu(5,7-Cl-8-HQ)** **113**	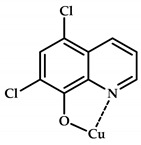	3.22	5.3	4.7
**Cu(CQ)** **114**	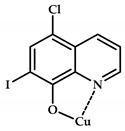	3.50	8.9	9.0
**Cu(5,7-I-8-HQ)** **115**	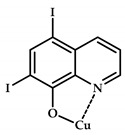	3.75	12.8	16.2

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
