# Peer review of "Copper Coordination Compounds as Biologically Active Agents"

_ijms, 2020, doi:10.3390/ijms21113965_

Round 1

Reviewer 1 Report

The paper gives an overview of some of the copper complexes reported over the last few years and endowed with biological activity. Actually, several review articles dedicated to copper complexes for medicinal applications have been published. On the other hand, the rapid development of this research field does justify the proliferation of such review articles, which provide a continuously up-to-date vision of the research forefront for medicinal inorganic chemists.

For this reason, this contribution can be accepted for publication in the International Journal of Molecular Sciences. However the manuscript has some weak points to be addressed before being published.

The first point that authors should better clarify, concerns the criteria on which their survey is based. In the Introduction it’s reported: “This review is devoted to recent developments in the use of copper coordination compounds as drugs with multiple therapeutic effects” . From my first quick reading, I understood from this sentence that the paper would be related to “copper coordination compounds as drugs with multiple therapeutic effects” i.e. to those compounds which exert multiple therapeutic effects, and this would have been an original approach, but then I realized that this is not the case as many complexes endowed only with antitumoral activity have been reviewed.

The survey is not exhaustive of the last years (e.g. in 2018 more than 100 papers have been published concerning only the antitumoral activity of copper complexes and not all are reported), but authors have used some selection criteria which should be clearer to the reader. Also the range of time considered (the last 3 years?) should be better specified.

For example, it’s not clear why Cu(I) compounds in particular Cu(I) phosphino derivatives have been completely neglected. The noticeable antiproliferative activity of phosphino derivatives has been largely demonstrated and many studies on their mechanism(s) of action have been reported. Their chemistry and biological activity have been reviewed in 3 chapters (chapters 3, 4 and 5) of a recently published book : “Copper(I) Chemistry of Phosphines, Functionalized Phosphines and Phosphorus Heterocycles” Ed. Maravanji S. Balakrishna ISBN: 978-0-12-815052-8, 2019, Elsevier.

Some phosphino complexes have also demonstrated antibacterial or antimalarial activity (e.g. New J. Chem., 2019, 43, 19318 ; Journal of Inorganic Biochemistry 166 (2017) 1–4).

In my opinion the rationale behind the choice of the cited papers has to be better clarified.

The conclusion is too general. In fact, as the review concerns different applications of copper coordination complexes, authors should summarize the most promising results for each applications (antitumor, antimicrobial, etc.) and also the proposed mechanisms and targets for each biological activity trying to evidence any possible SARs. This would be very helpful for the readers. Hopefully, by a critical analysis of the reported data, they should try to indicate the new directions of the research to scientists working in this field.

The paper is clear, well written and well organized, but , as reported below, all references and their exact numbering in the text must be checked.

Other points:

  • Introduction: It would be worth mentioning some more recent reviews such as for example:

“Progress in Copper Complexes as Anticancer Agents” Tabti et al., Med Chem (Los Angeles) 2017, 7:5 DOI: 10.4172/2161-0444.1000445 (this paper reports many in-vivo data); “A Perspective – can copper complexes be developed as a novel class of therapeutics? Mohamed Wehbe et al. , Dalton Trans. 2017 and also “Metal Compounds against Neglected Tropical Diseases” Yih Ching Ong, et al. Chem. Rev. 2019, 119, 730−796 where the antileshmanial activity of several Cu complexes is reviewed.

  • Page 3 – Line 89 the reagent is not diphenol, but 5-dimethylcyclohexane-1,3-dione.
  • 34: pages are 619-638 not 629- 638.
  • Figure 1: compound 1 and 2?
  • Page 14 – Line 337 compound 44 should be better reported in chapter 2.5
  • Page 17 - Line 397 ref. 58 is wrong: the right number is 65. Hence p.18 line 424 ref. 65 becomes 66, p.19 line 439 Majouga et. al. becomes ref. 67 instead of 66. The correspondence between references and numbering must be carefully checked throughout the text.
  • 61 : year is 2019 not 2018
  • 68: volume is 252 not 2.
  • Superscript and subscript must be formatted throughout the manuscript (IC50, 64Cu etc.)
  • Page 25- Lines 633-637 : It would be better to always use the same notation fenoprofen (fen) instead of phenoprofen (phen).
  • A list of the abbreviations and of the acronyms of cancer cell lines cited in the review would be useful.
  • Among complexes containing an active ligand (several of recent ones have not been taken into account), natural products should also be considered. Interesting papers have been recently published. Here I report only a couple of examples: Dalton Trans., 2019, 48, 15646–15656 “Biological evaluation of optically pure chiral binuclear copper(II) complexes based on a rosin derivative as highly potential anticancer agents Bao-Li Fei et al.”; European Journal of Pharmacology 815 (2017) 147–155 “Hinokitiol copper complex inhibits proteasomal deubiquitination and induces paraptosis-like cell death in human cancer cells” Xin Chen et al..

Author Response

Dear reviewer, thank you for your time and appreciation of our work. We tried to take into account all your comments, which increased the quality of our review. We have answered each of your comments below.

The paper gives an overview of some of the copper complexes reported over the last few years and endowed with biological activity. Actually, several review articles dedicated to copper complexes for medicinal applications have been published. On the other hand, the rapid development of this research field does justify the proliferation of such review articles, which provide a continuously up-to-date vision of the research forefront for medicinal inorganic chemists.

For this reason, this contribution can be accepted for publication in the International Journal of Molecular Sciences. However the manuscript has some weak points to be addressed before being published.

The first point that authors should better clarify, concerns the criteria on which their survey is based. In the Introduction it’s reported: “This review is devoted to recent developments in the use of copper coordination compounds as drugs with multiple therapeutic effects” . From my first quick reading, I understood from this sentence that the paper would be related to “copper coordination compounds as drugs with multiple therapeutic effects” i.e. to those compounds which exert multiple therapeutic effects, and this would have been an original approach, but then I realized that this is not the case as many complexes endowed only with antitumoral activity have been reviewed. - multiple” was replaced by “various” in order to avoid incompletement.

The survey is not exhaustive of the last years (e.g. in 2018 more than 100 papers have been published concerning only the antitumoral activity of copper complexes and not all are reported), but authors have used some selection criteria which should be clearer to the reader. Also the range of time considered (the last 3 years?) should be better specified. In my opinion the rationale behind the choice of the cited papers has to be better clarified.

In this review we summed up different publications of recent years, paying attention to the variety of biological studies that allow us to assess the potential of coordination copper compounds. This review will be useful to researchers engaged in the search for the mechanism of action of copper-containing antitumor agents.

There are indeed a lot of publications devoted to new coordination compounds of copper, and it is difficult to cover all of them even in the last three years. We tried to mention both classic articles devoted to cytotoxic agents, and versatile studies of the mechanism of action, recent clinical studies. Also, in this review, we examined not only antitumor drugs, but also the use in other areas, copper-containing coordination compounds as PET-visualizers of amyloids, markers of hypoxia, antibiotic agents, antituberculous agents, NSAID drugs.

For example, it’s not clear why Cu(I) compounds in particular Cu(I) phosphino derivatives have been completely neglected. The noticeable antiproliferative activity of phosphino derivatives has been largely demonstrated and many studies on their mechanism(s) of action have been reported. Their chemistry and biological activity have been reviewed in 3 chapters (chapters 3, 4 and 5) of a recently published book : “Copper(I) Chemistry of Phosphines, Functionalized Phosphines and Phosphorus Heterocycles” Ed. Maravanji S. Balakrishna ISBN: 978-0-12-815052-8, 2019, Elsevier.

Some phosphino complexes have also demonstrated antibacterial or antimalarial activity (e.g. New J. Chem., 2019, 43, 19318 ; Journal of Inorganic Biochemistry 166 (2017) 1–4) - We have added an additional chapter with the above publications.

The conclusion is too general. In fact, as the review concerns different applications of copper coordination complexes, authors should summarize the most promising results for each applications (antitumor, antimicrobial, etc.) and also the proposed mechanisms and targets for each biological activity trying to evidence any possible SARs. This would be very helpful for the readers. Hopefully, by a critical analysis of the reported data, they should try to indicate the new directions of the research to scientists working in this field. – we have corrected conclusions.

The paper is clear, well written and well organized, but , as reported below, all references and their exact numbering in the text must be checked.

Other points:

  • Introduction: It would be worth mentioning some more recent reviews such as for example:

“Progress in Copper Complexes as Anticancer Agents” Tabti et al., Med Chem (Los Angeles) 2017, 7:5 DOI: 10.4172/2161-0444.1000445 (this paper reports many in-vivo data); “A Perspective – can copper complexes be developed as a novel class of therapeutics? Mohamed Wehbe et al. , Dalton Trans. 2017 and also “Metal Compounds against Neglected Tropical Diseases” Yih Ching Ong, et al. Chem. Rev. 2019, 119, 730−796 where the antileshmanial activity of several Cu complexes is reviewed. - We have added citation of several reviews in the introduction.

  • Page 3 – Line 89 the reagent is not diphenol, but 5-dimethylcyclohexane-1,3-dione.
  • 34: pages are 619-638 not 629- 638.-
  • Figure 1: compound 1 and 2? – we have added the number of compounds.
  • Page 14 – Line 337 compound 44 should be better reported in chapter 2.5 –
  • Page 17 - Line 397 ref. 58 is wrong: the right number is 65. Hence p.18 line 424 ref. 65 becomes 66, p.19 line 439 Majouga et. al. becomes ref. 67 instead of 66. The correspondence between references and numbering must be carefully checked throughout the text. – we have corrected this mistake and checked all references carefully.
  • 61 : year is 2019 not 2018
  • 68: volume is 252 not 2.
  • Superscript and subscript must be formatted throughout the manuscript (IC5064Cu etc.)
  • Page 25- Lines 633-637 : It would be better to always use the same notation fenoprofen (fen) instead of phenoprofen (phen).
  • A list of the abbreviations and of the acronyms of cancer cell lines cited in the review would be useful.
    we added an acronyms of all cell lines to the main text.
  • Among complexes containing an active ligand (several of recent ones have not been taken into account), natural products should also be considered. Interesting papers have been recently published. Here I report only a couple of examples: Dalton Trans., 2019, 48, 15646–15656 “Biological evaluation of optically pure chiral binuclear copper(II) complexes based on a rosin derivative as highly potential anticancer agents Bao-Li Fei et al.”; European Journal of Pharmacology 815 (2017) 147–155 “Hinokitiol copper complex inhibits proteasomal deubiquitination and induces paraptosis-like cell death in human cancer cells” Xin Chen et al. – we have added an additional chapter devoted to natural products-based coordination compounds. Recently, a review devoted to natural products-based copper coordination compounds was published, which we also cited. [Anticancer Agents Med Chem.2019;19(1):48-65].

Reviewer 2 Report

The manuscript by Krasnovskaya et al. describes well recent advances in Cu-coordination compounds which can potentially used as drugs against wide variety of diseases including cancer, neurodegenerative disease, malaria etc. Moreover, some of the agents described could also be used as PET imaging agents. The article is well structured and each section gives good overview of the current status in the Cu-coordinated drugs development by covering many recent articles on this topic. Overall, I believe that presented review could serve as a good overview of the state of the art for scientist working in this field.

However, while reading the manuscript, there were few points that I would like to point out.

1.) Tables 1, 6, 8, and 9 and Lines 139-141: Accuracy of standard deviation should be the same as the number of decimal places and ± sign should be consistently used with or without space before and after it but not with different styles as it is now.

2.) Figure 1: Compounds should be also marked as 1 and 2, both on figure and in figure caption. Moreover, namings for those two compounds has two different styles which is a bit confusing.

3.) Figures 4 and 40: Figure of X-ray structure is of poor quality and namings of Cu atoms can be barely seen.

4.) Lines 170 and 171: What is written here "the effectiveness of coordination compound 13 in combination with rifampicin as an antituberculosis agent" is not in agreement with the caption of Figure 7.

5.) Figures 11, 27, and 29: Size scale in each panel of the figure is not readable.

6.) Figure 14: What is the meaning of "Tissue-1" in panel C? And 65Cu/63Cu should be written with 65 and 63 in superscript. 

7.) Figure 30: I believe that the equation for the second chemical reaction is wrong.

8.) Figure 38: It would help the reader to rotate the right figure for 90 degrees counter clockwise.

9.) Figure 22: I do not understand the meaning of numbers in parentheses. Moreover, the figure is of poor quality as it is Figure 25.

10.) Subtitles 2.X: Please be consistent, sometimes it is written in italic and sometimes in bold. Moreover, sometimes coma is used between the two Cu-coordinating atoms and sometins slash.

11.) Always write isotope numbers in superscript before the atom name (Lines 235, 268-270)

12.) Lines 241-246: It is same as paragraph above.

13.) Lines 275, 288: Check correct usage of capital letters.

14.) Lines 298 and 314: Staphylococcus aureus should be written in italic.

15.) Line 309: "are" should not be in bold

16.) References 30 and 67 have different style than others.

17.) Figure 46: Compound naming is missing on the figure.

18.) Finally, I would advise authors to use et al. (instead of et. al.), to write latin expression, such as "in vivo", "in vitro", "ex vivo", and "in situ" in italic as it was correctly written in Line 570. Compounds naming should always be in bold, both in text and in Figures (Lines 100, 102, 115, 125-127, 131, 569, Table 7, and Figure 23. Moreover, font style should be consistently used throughout the manuscript and not as it is now, where many different fonts are used, specially in the figure captions.

Author Response

Dear reviewer, thank you for your time and appreciation of our work. We tried to take into account all your comments, which increased the quality of our review. We have answered each of your comments below.

The manuscript by Krasnovskaya et al. describes well recent advances in Cu-coordination compounds which can potentially used as drugs against wide variety of diseases including cancer, neurodegenerative disease, malaria etc. Moreover, some of the agents described could also be used as PET imaging agents. The article is well structured and each section gives good overview of the current status in the Cu-coordinated drugs development by covering many recent articles on this topic. Overall, I believe that presented review could serve as a good overview of the state of the art for scientist working in this field.

However, while reading the manuscript, there were few points that I would like to point out.

  • Tables 1, 6, 8, and 9 and Lines 139-141: Accuracy of standard deviation should be the same as the number of decimal places and ± sign should be consistently used with or without space before and after it but not with different styles as it is now. - We checked the standard deviations in the original article, unfortunately, the authors cited them in this format.
  • Figure 1: Compounds should be also marked as 1 and 2, both on figure and in figure caption. Moreover, namings for those two compounds has two different styles which is a bit confusing.
  • Figures 4 and 40: Figure of X-ray structure is of poor quality and namings of Cu atoms can be barely seen.Unfortunately, the X-ray data are presented in the original article in this form.

4.) Lines 170 and 171: What is written here "the effectiveness of coordination compound 13 in combination with rifampicin as an antituberculosis agent" is not in agreement with the caption of Figure 7. We checked the data with original article and corrected the mistake.

  • Figures 11, 27, and 29: Size scale in each panel of the figure is not readable.

6.) Figure 14: What is the meaning of "Tissue-1" in panel C? And 65Cu/63Cu should be written with 65 and 63 in superscript. We checked the data with original article and corrected this picture.

7.) Figure 30: I believe that the equation for the second chemical reaction is wrong.  We checked the data with original article and corrected the mistake.

8.) Figure 38: It would help the reader to rotate the right figure for 90 degrees counter clockwise. Done.

9.) Figure 22: I do not understand the meaning of numbers in parentheses. Moreover, the figure is of poor quality as it is Figure 25.We have added the subscriptions.

10.) Subtitles 2.X: Please be consistent, sometimes it is written in italic and sometimes in bold. Moreover, sometimes coma is used between the two Cu-coordinating atoms and sometins slash. Corrected.

11.) Always write isotope numbers in superscript before the atom name (Lines 235, 268-270) Corrected.

12.) Lines 241-246: It is same as paragraph above. Corrected.

13.) Lines 275, 288: Check correct usage of capital letters. Corrected.

14.) Lines 298 and 314: Staphylococcus aureus should be written in italic. Corrected.

15.) Line 309: "are" should not be in bold Corrected.

16.) References 30 and 67 have different style than others. Corrected.

17.) Figure 46: Compound naming is missing on the figure. Corrected.

18.) Finally, I would advise authors to use et al. (instead of et. al.), to write latin expression, such as "in vivo", "in vitro", "ex vivo", and "in situ" in italic as it was correctly written in Line 570. Corrected.

Compounds naming should always be in bold, both in text and in Figures (Lines 100, 102, 115, 125-127, 131, 569, Table 7, and Figure 23. Corrected.

Moreover, font style should be consistently used throughout the manuscript and not as it is now, where many different fonts are used, specially in the figure captions. We have applied a Palantino Linotype font to all pictures and schemes.